# Fast and accurate edge orientation processing during object manipulation

**J Andrew Pruszynski[1,2,3,4,5]\*, J Randall Flanagan[6,7], Roland S Johansson[5]**

[1]Department of Physiology and Pharmacology, Western University, London, Canada; [2]Department of Psychology, Western University, London, Canada; [3]Robarts Research Institute, Western University, London, Canada; [4]Brain and Mind Institute, Western University, London, Canada; [5]Department of Integrative Medical Biology, Umea University, Umea, Sweden; [6]Centre for Neuroscience Studies, Queen's University, Kingston, Canada; [7]Department of Psychology, Queen's University, Kingston, Canada

**Abstract** Quickly and accurately extracting information about a touched object's orientation is a critical aspect of dexterous object manipulation. However, the speed and acuity of tactile edge orientation processing with respect to the fingertips as reported in previous perceptual studies appear inadequate in these respects. Here we directly establish the tactile system's capacity to process edge-orientation information during dexterous manipulation. Participants extracted tactile information about edge orientation very quickly, using it within 200 ms of first touching the object. Participants were also strikingly accurate. With edges spanning the entire fingertip, edge-orientation resolution was better than 3° in our object manipulation task, which is several times better than reported in previous perceptual studies. Performance remained impressive even with edges as short as 2 mm, consistent with our ability to precisely manipulate very small objects. Taken together, our results radically redefine the spatial processing capacity of the tactile system.
DOI: https://doi.org/10.7554/eLife.31200.001

**\*For correspondence:**
andrew.pruszynski@uwo.ca

**Competing interests:** The authors declare that no competing interests exist.

## Introduction

Putting on a necklace involves holding open a clasp while aligning it with a ring, a process that requires quickly and accurately determining and controlling each object's orientation. In this and many other fine manipulation tasks, information about an object's orientation is based largely on how its edges activate mechanoreceptors in the glabrous skin of the fingertips. Indeed, fingertip numbness due to events like cold exposure and nerve injury can degrade or even preclude fine manual dexterity (*Moberg, 1958*; *Chemnitz et al., 2013*).

No previous studies have examined the speed and accuracy with which the neural system extracts and expresses tactile edge orientation information during object manipulation tasks that require fine manual dexterity. However, perceptual studies of tactile edge orientation have been done (*Lechelt, 1992*; *Bensmaia et al., 2008*; *Peters et al., 2015*), and for edges that span a large portion of the fingertip, the reported orientation acuity is 10–20°. For shorter edges, which only engage a small part of the fingertip, as is typical during fine manipulation tasks like buttoning, the reported perceptual orientation acuity is even cruder, around 90° for a 2 mm long edge (*Peters et al., 2015*). These psychophysical measures appear too crude to underlie the control of dexterous object manipulation and tell little about the speed by which the brain can extract and use tactile edge orientation information.

Here, we used a novel experimental paradigm to establish the tactile system's ability to process edge-orientation information during object manipulation. In our main experiment, participants used their fingertip to contact a randomly oriented dial and, based on only tactile information gathered

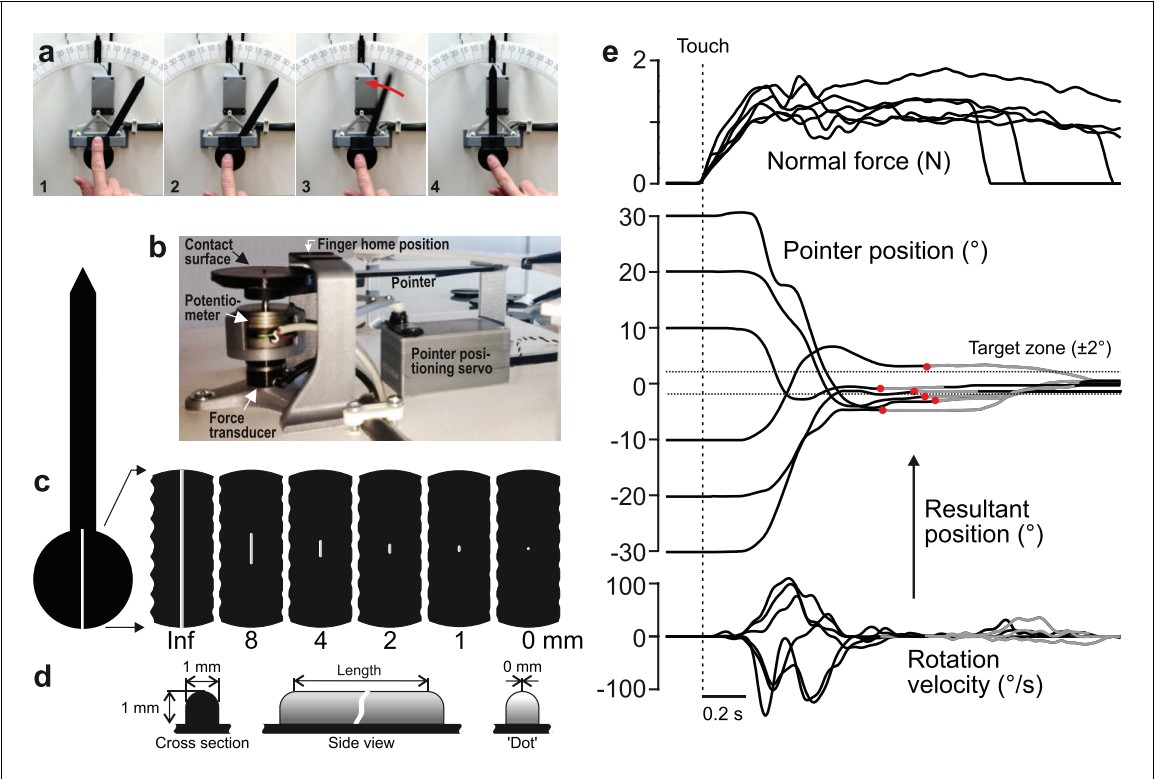

**Figure 1.** Experimental approach. (**a**) Four principle phases of the pointer-alignment trials. (**b**) Photograph of the apparatus. (**c**) The left panel shows a top-down schematic view of the dial and pointer along with an exemplar fingerprint superimposed on the contact surface for scale purposes. The six panels on the right show the six edge lengths. The edge that spanned the entire area contacted by fingertip was termed the infinite edge and the 0 mm edge refers to raised dot stimulus. (**d**) Cross-sectional and side views of the edges. (**e**) Normal force, pointer position and rotation velocity shown for six superimposed exemplar trials with the six initial dial orientations. Data aligned on initial touch (vertical line). Dashed horizontal lines represents the target ±2° zone. The resultant pointer position was measured when the rotation velocity fell below 10°/s (red dots). Gray segments of the traces represent final adjustments of the orientation with the shutter glasses opened to allow visual guidance of the movement during the final adjustment of the pointer into the target zone when required.

DOI: https://doi.org/10.7554/eLife.31200.003

The following figure supplement is available for figure 1:

**Figure supplement 1.** Participants quickly learned the tactile pointer-alignment task.

DOI: https://doi.org/10.7554/eLife.31200.004

from a raised edge located on the dial, quickly rotated the dial to orient a pointer towards a target position (*Figure 1a–d*). We found that participants oriented the pointer strikingly well. On average, participants were within 3° of the target orientation for edges spanning the entire contact area of the fingertip, similar to their performance in a visually-guided version of the same task, and considerably better than the acuity of edge orientation processing previously reported in studies of tactile perception (*Lechelt, 1992*; *Bensmaia et al., 2008*; *Peters et al., 2015*). Performance remained impressive even with much shorter edges, with participants orienting the unseen dial to within 11° of the target orientation for a 2 mm long edge. We also found that participants gathered and processed the relevant tactile information quickly, initiating appropriate dial rotation within ~200 ms of initially touching the edge. Based on a simple model, we propose that this exquisite capacity may reflect a previously largely overlooked feature of the peripheral tactile apparatus (*Friedman et al., 2002*; *Dodson et al., 1998*; *Wheat et al., 1995*; *Saal et al., 2017*) – namely, that first-order tactile neurons branch in the fingertip skin (*Cauna, 1956*; *Cauna, 1959*; *Nolano et al., 2003*) and have cutaneous receptive fields with multiple highly-sensitive zones (or 'subfields') (*Johansson, 1978*; *Phillips et al., 1992*; *Pruszynski and Johansson, 2014*; *Suresh et al., 2016*).

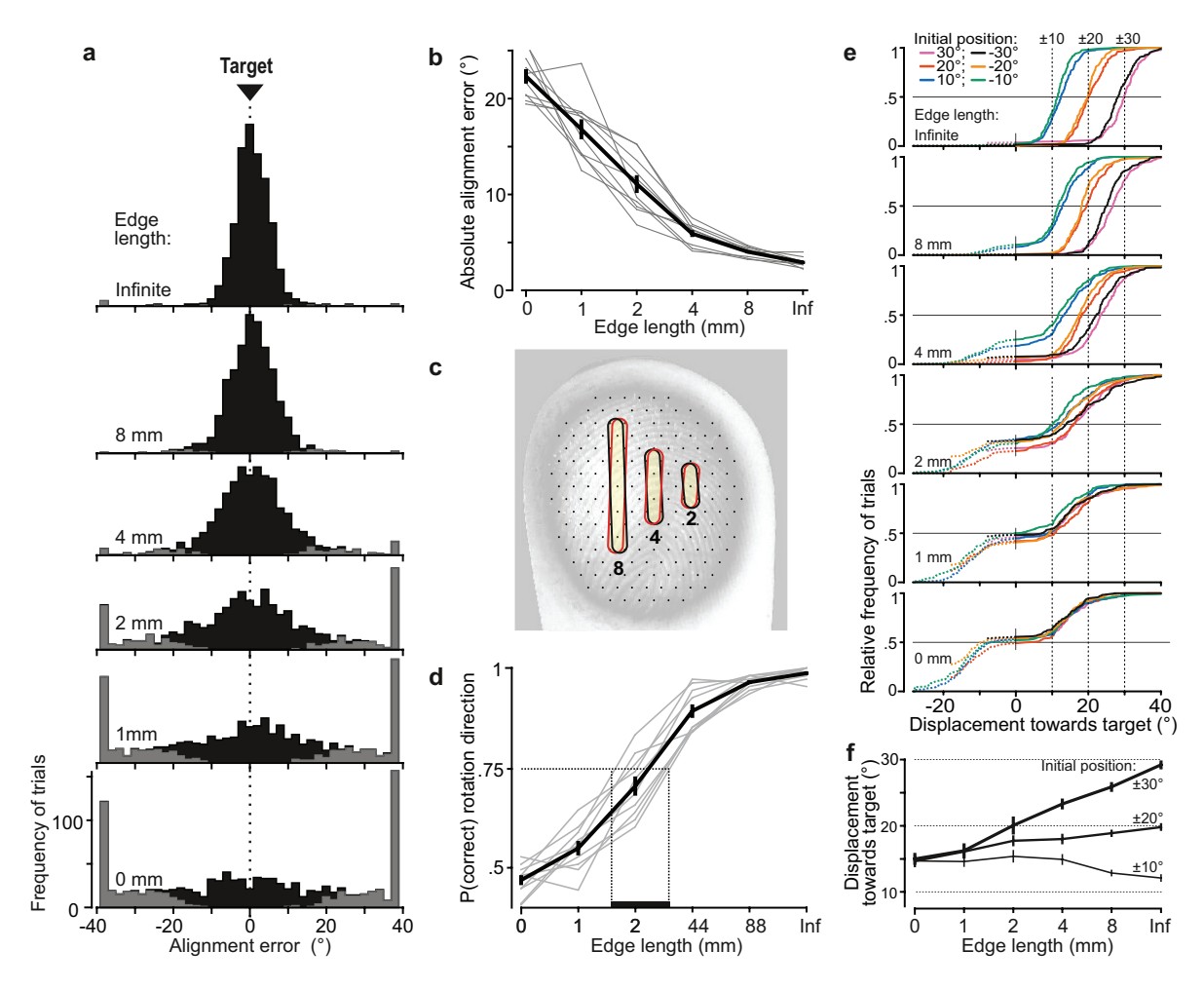

**Figure 2.** Alignment accuracy during tactile pointer-alignment trials. (a) Distribution of the alignment error for the various edge lengths for all trials by all ten participants (108 trials/participant and edge length). Gray segments of the distributions refer to trials with rotation in the wrong direction. The accumulation of data at the ±38° represents trials in which the pointer reached the end of its movement range (see Materials and methods). (b) Absolute alignment error (deviation from the 0° target position) as a function of edge length based on median values for individual subjects (gray lines) and the corresponding data averaged across participants (black line; means ±1 sem). (c) Contours superimposed on a fingerprint – photographed through a flat glass plate – show the 8, 4 and 2 mm edges twice with an orientation difference that corresponds to the average alignment errors with these edges. For reference, superimpose of the fingertip is an array of black dots, laid out in a hexagonal array with a center-to center spacing of 1 mm, which approximately correspond to the spacing of receptive field centers of relevant tactile neurons if uniformly spaced across the fingertip. (d) Proportion of trials with rotations in the correct direction as function of edge length for each participant for all initial dial orientations pooled (gray lines) and the corresponding data averaged across participants (black line). Under the criterion that 75% correct responses define the threshold level, the vertical dashed lines indicates an estimation of the range across participants of threshold of edge length for correct rotation direction. (e) Cumulative frequency distribution of the pointer displacement referenced to movement in the direction of the target for trials performed by all participants with each edge length and initial dial orientation. The vertical dashed lines indicate the displacement required to reach the target position. The dashed segments of the distributions refer to trials with rotation in incorrect direction (i.e., negative displacement values) and are curtailed by the pointer reaching the end of its movement range. (f) Pointer displacement in the correct direction as a function of initial dial orientation and edge length shown as mean values across subjects (±1 sem; N = 10) based on participants' medians. The dashed horizontal lines indicate the displacement required reaching the target for the 10, 20 and ±30° initial dial orientations. Data are pooled across the 10, 20 and ±30° orientations since there was no significant effect of sign of the orientation on the pointer displacements these initial orientations.

DOI: https://doi.org/10.7554/eLife.31200.005

The following source data is available for figure 2:

**Source data 1.** Underlying data points for *Figure 2b, d and f*.
DOI: https://doi.org/10.7554/eLife.31200.006

# Results

In our main experiment, ten study participants stood at a table holding the tip of their right index finger at a home position located above a dial (*Figure 1a–1*). An auditory signal instructed the participants to execute the task, which was to move their finger down from the home position to contact the dial at its center of rotation (*Figures 1a–2*) and, based on tactile information gathered from a raised edge located on the dial (see below), to rotate the dial and orient a pointer, attached to the dial, from its initial position towards a center position (*Figures 1a–3*). This action corresponds to touching, from above, the needle of a compass and, by rotating the fingertip, orienting it from some initial position, say northwest or northeast, to due north (labeled 0°). The initial orientation of the dial was randomized across the trials yielding six initial pointer positions relative to the due north target (30, 20, ±10°). Hence, correctly orienting the dial required rotating the dial in the direction opposite the initial orientation, either clockwise or counter-clockwise by 10, 20 or 30°. Shutter glasses prevented the participants from seeing the dial and pointer before and during the rotation. When the dial rotation ended, we measured the resultant pointer position and assessed the alignment error from due north (*Figures 1a–4*, *Video 1*). At the same time, the shutter glasses opened, which gave the participant visual feedback about their performance. If the resultant pointer position was off the due north target by more than ±2°, participants were required to adjust, under visual guidance, the pointer to within ±2° of the due north target position. After target acquisition, the participant returned to the home position where, between trials, their fingertip rested on a horizontal plate with a raised edge in its center pointing towards the target position (i.e., 0°). This edge offered participants a tactile reference for the finger's home position and may have helped participants maintain a representation of the direction to the target position. A raised edge on the contacted surface, the length of which constituted a key experimental variable, was oriented in the direction of the dial's pointer and provided tactile information about the dial's orientation relative to the fingertip (*Figure 1b–d*). *Figure 1e* shows exemplar pointer-alignment trials from one participant. When the participant contacted the dial, the normal force increased to a plateau-like force that was maintained until the trial ended. Typically, the rotation of the pointer started while the contact force was still increasing. The rotation velocity profile often showed one major velocity peak, but could also show two or more peaks indicating that one rotation could sometimes comprise two or even more sub-movements.

## Tactile edge orientation is extracted and processed very accurately in manipulation

Participants learned the tactile pointer-alignment task quickly during a practice block, and there were no signs of further learning during the experiment (*Figure 1—figure supplement 1*). *Figure 2a* shows the distribution of alignment errors for all pointer-alignment trials by all ten participants separated for each of the six edge lengths, ranging from a small dot of zero length that provided no orientation information to an infinite edge spanning the entire area contacted by the fingertip (*Figure 1c*). For the infinite edge, the resulting pointer positions were concentrated around the 0° target position. As the edge length decreased, the distribution gradually became broader indicating that, on average, the alignment error increased. An increased frequency of trials with rotation in the wrong direction, that is, away from the target, contributed to this increase (gray segments of the distributions in *Figure 2a*).

*Figure 2b* shows the absolute value of the alignment error for all trials (correct and incorrect rotation directions) as a function of edge length based on median values for individual participants. Edge length significantly affected the absolute alignment error ($F_{5,45}$ = 238.5, $p < 10^{-6}$), which gradually decreased with increasing length. With the infinite edge, the error was 2.9 ± 0.5° (mean ±1 SD across participants) and with the 2 mm edge it was 11.1 ± 2.9°, which was about one half of the error with the raised dot (i.e., 0 mm edge length) representing chance performance. *Figure 2c* illustrates how the sensitivity to edge orientation relates to the events at the fingertip by illustrating the 2, 4 and 8 mm edges projected twice on a fingerprint at an angular difference of 4.0, 5.9 and 11.1°, respectively. These angular differences correspond to the average absolute alignment error with these edge lengths, and result in positional changes at the end of the edge of 0.28, 0.21 and 0.19 mm, respectively, if rotated around their centers.

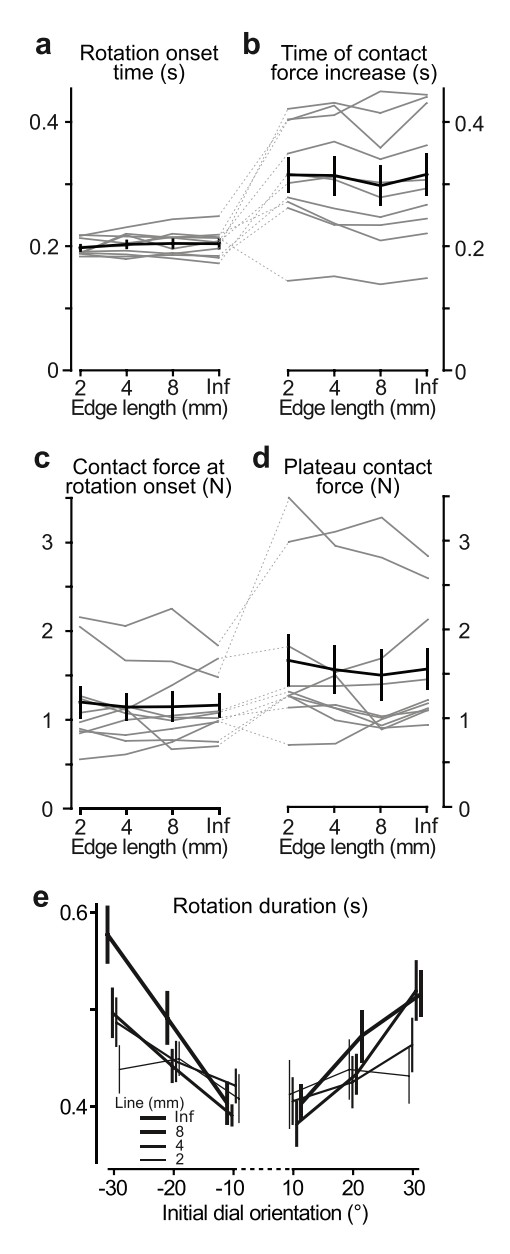

**Figure 3.** Contact behavior and temporal parameters in tactile pointer-alignment trials. (**a,b**) Time of onset of the orienting of the dial ('Rotation onset time') and the time when the contact force reached its plateau-like state ('Time of contact force increase') as a function of edge length referenced to the time of initial touch of dial. (**c,d**) Contact force at the time of the start of dial rotation and during the plateau-like state of the force, respectively. (**a–d**) Gray lines indicate median values for individual subjects and black line represents their mean values averaged across participants. Error bars indicate the standard error of the mean. (**e**) The duration of the dial rotation as a function of the dial's initial orientation for each of the edges that were 2 mm and longer. Lines indicate means across participants' medians. Error bars indicate the standard error of the mean.

DOI: https://doi.org/10.7554/eLife.31200.007

The following source data and figure supplement are available for figure 3:

**Source data 1.** Underlying data points for *Figure 3a–e*.
DOI: https://doi.org/10.7554/eLife.31200.009

**Figure supplement 1.** Presence of sub-movements did not influence alignment accuracy or direction errors in the tactile pointer-alignment task.
DOI: https://doi.org/10.7554/eLife.31200.008

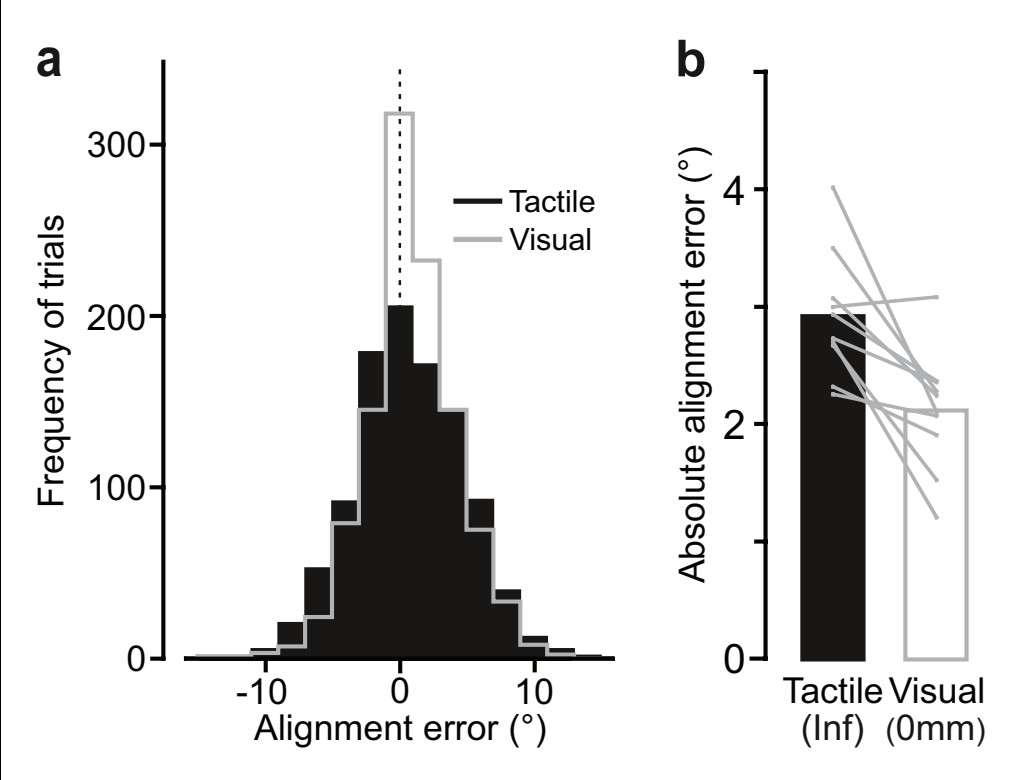

**Figure 4.** Comparing performance in the visual and the tactile pointer-alignment tasks. (**a**) Distribution of the alignment error during the visual (gray) and tactile (black) pointer-alignment tasks for all trials by all ten participants (108 trials/participant and task). (**b**) Absolute alignment error in the two tasks. Height of black and white bars indicates mean values across participants' medians in the tactile and visual condition, respectively, and gray lines indicate median values for each participant and condition.

DOI: https://doi.org/10.7554/eLife.31200.010
The following source data and figure supplement are available for figure 4:

**Source data 1.** Underlying data points for **Figure 4b**.
DOI: https://doi.org/10.7554/eLife.31200.012
**Figure supplement 1.** Comparing performance in the visual and tactile pointer-alignment tasks.
DOI: https://doi.org/10.7554/eLife.31200.011

One reason that alignment error increased with shorter edges was that participants more frequently rotated the dial in the wrong direction ($F_{5,45} = 258.4$; $p<10^{-6}$). The proportion of movements in the correct direction gradually decreased from nearly 100% with the infinite edge down to chance performance (~50%) with the raised dot (**Figure 2c**). If 75% correct responses define threshold performance, as is common in two alternative forced choice (2AFC) tasks, the average threshold of edge length for correct rotation direction was around 2 mm.

Another reason for the increased alignment error with shorter edges was that the scaling of pointer displacement based on the initial dial orientation became poorer for trials in the correct direction. **Figure 2e** shows, for each initial dial orientation and edge length, the distribution of pointer displacements in the direction of the target for all trials by all participants (negative displacements indicate movements in the incorrect direction) and **Figure 2f** shows the displacement for movements in the correct direction based on participants' medians. With the infinite edge, participants appropriately scaled pointer displacements in the sense that the alignment error was, on average, close to zero for each initial orientation (top panel in **Figure 2e**). However, there was a tendency to undershoot the target with the ±30° initial orientations and overshoot the target with ±10° initial orientations. When the edge length decreased, for movements in the correct direction participants tended to increasingly undershoot the target for the 30 and ±20° initial orientations, whereas they tended to overshoot with the ±10° orientations (**Figure 2f**). Indeed, there was a

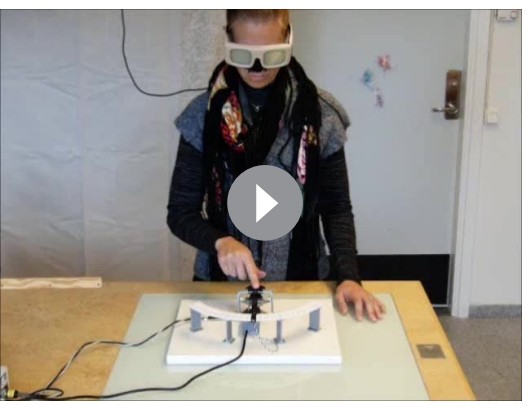

**Video 1.** Our apparatus and a few sample trials.
DOI: https://doi.org/10.7554/eLife.31200.013

significant interaction between edge length and initial orientation ($F_{25,225}$ = 21.1; $p<10^{-6}$) together with main effects of edge length ($F_{5,45}$ = 14.1; $p<10^{-6}$) and initial orientation ($F_{5,45}$ = 77.5; $p<10^{-6}$) on the displacement in the correct direction. Post-hoc analyses failed to show a significant effect of sign of the initial orientation on the pointer displacement in the correct direction for the 10, 20 and ±30° orientations. For the raised dot, which provided no edge orientation information, participants nevertheless generated pointer movements of ~15°. However, since these were in one direction or the other, with approximately equally probability and amplitude, virtually no pointer displacement occurred on average (bottom panel in *Figure 2e*). Performance with the 1 mm edge was similar to that observed with the raised dot although some sensitivity to the initial dial orientation was apparent. For the 4 and 8 mm edges, we noted that the proportion of trials with movements in the wrong direction tended to be greater for the ±10° than for the 20 and ±30° initial orientations (*Figure 2e*). This impression was statistically supported by an interaction effect of initial orientation and edge length ($F_{25,225}$ = 3.0; $p=8\times10^{-6}$) on the proportion of movement in the correct direction, along with a main effect of the initial orientation ($F_{5,45}$ = 8.3; $p=10^{-5}$).

Because the dial was initially oriented at one of six orientations in the main experiment, it is possible that participants may have learned six responses and then selected one of these responses based on coarse discrimination among the six initial orientations (10° apart). There are at least two reasons why this situation is unlikely. First, participants showed no tendency to move in multiples of 10° with short edges suggesting that they utilized tactile information about dial orientation in an analog manner to program the movement rather than attempting to categorize which of the six possible orientations was presented and then selecting the appropriate motion (*Figure 2e*). Second, in a follow-up experiment performed with the infinite edge and involving 50 rather than six initial dial orientations (see Materials and methods), the absolute alignment (2.7 ± 0.4°; mean ±1 SD across participants) did not differ significantly from that recorded in the main experiment (2.9 ± 0.5°; $F_{1,18}$ = 1.19; $p=0.29$).

Taken together, we found that tactile information about edge orientation could effectively guide manipulation for edges that were 2 mm and longer and, with an edge of infinite length relative to the fingertip, alignment accuracy was, on average, better than 3°. We focused our remaining analyses on edges that were 2 mm and longer.

## Tactile edge orientation is extracted and processed very quickly in manipulation

Manual dexterity depends not only on access to accurate spatial tactile information but also requires that it is quickly available. We investigated how quickly participants extracted and used tactile edge orientation information in our pointer-alignment task by examining the time between initial contact with the dial and the onset of the rotation as well as the development of contact force and rotation kinematics.

Averaged across participants' medians, the time between touch and rotation onset was 0.20 ± 0.02 s (*Figure 3a*). Rotation onset typically occurred while the contact force was still increasing towards its plateau-like state (*Figures 1b* and *3b*), which, on average, was reached 0.31 ± 0.09 s after initial contact with the dial. Accordingly, the contact force at rotation onset (1.15 ± 0.44 N, *Figure 3c*) was typically smaller than the plateau force (1.64 ± 0.83 N, *Figure 3d*; $F_{1,9}$ = 11.5; $p=0.008$). Edge length and initial dial orientation showed no statistically significant effect on any of these measures.

The duration of dial rotation tended to increase with the required rotation amplitude but the size of this effect depended on the edge length. Shorter edges that yielded smaller pointer displacements also yielded shorter rotation durations (*Figure 3e*). This was reflected statistically as main

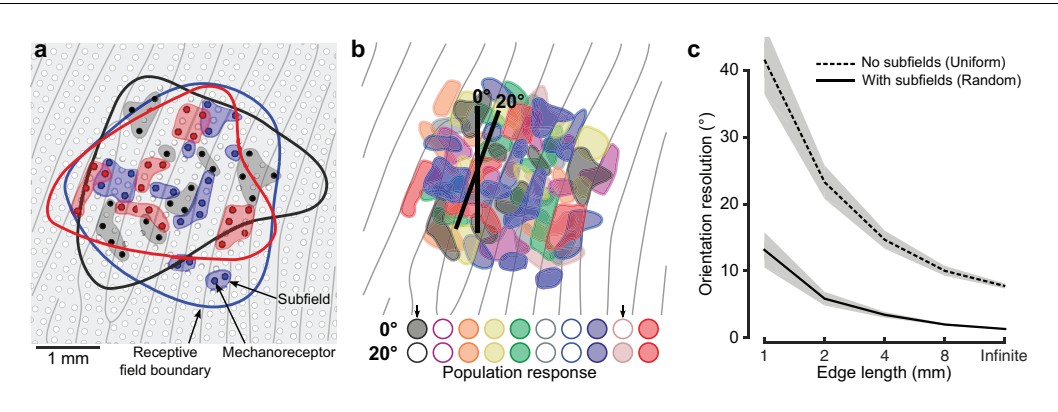

**Figure 5.** Neural mechanisms for edge orientation processing. (**a**) Schematic of a 5 × 5 mm square area on the skin surface. The gray lines and circles represent papillary ridges and mechanoreceptive end organs, respectively. Three colors of filled dots represent the mechanoreceptors (e.g. Meissner corpuscles) innervated by one of three first-order tactile neurons, the shaded area behind subsets of these mechanoreceptors represent subfields and the color-matched contour represents that neuron's receptive field boundary. (**b**) Top: Same format as (**a**) but showing color-coded subfields for 10 first-order tactile neurons. Note the high amount of receptive field overlap and subfield intermingling and that, in practice, even this representation is simplified as any point on the fingertip skin would activate ~36 of the relevant first-order tactile neurons (*Vallbo and Johansson, 1984*) (20 fast-adapting type 1: FA-1; 16 slow-adapting type 1: SA-1). The two edges (2 mm long) are superimposed on the layout are centered at the same location but differ in orientation by 20˚. Bottom: Activation pattern of the population of neurons in the cartoon above. Neurons are filled if the edge touches any of its subfields and unfilled otherwise. Arrows point to two neurons that change their state for the two edge orientations. (**c**) Output of our model, relating subfields to the neuronal populations' ability to signal edge orientation (ordinate) as a function of edge length (abscissa). Here we directly contrast two synthetic populations where: (1) each unit has a uniform receptive field by virtue of being connected to one receptive element the same size as its receptive field and (2) each unit has subfields by virtue of being connected to a random number (2–64) of receptor elements (each 250 µm in diameter). Each simulation was repeated 100 times for each edge length. The lines indicate the mean and the shaded areas represent the 95% confidence interval.

DOI: https://doi.org/10.7554/eLife.31200.014

The following source data and figure supplement are available for figure 5:

**Source data 1.** Underlying data points for *Figure 5c*.

DOI: https://doi.org/10.7554/eLife.31200.016

**Figure supplement 1.** Schematic and flow chart of discrimination model.

DOI: https://doi.org/10.7554/eLife.31200.015

effects of both edge length and initial dial orientation on the rotation duration ($F_{3,27}$ = 6.2, p=0.002 and $F_{5,45}$ = 15.0, p<$10^{-6}$, respectively) as well as an interaction between these factors ($F_{15,135}$ = 4.5, p<$10^{-6}$).

Although the task explicitly emphasized accuracy, participants initiated the rotation movement much sooner after contact (~0.2 s) than the maximum permitted delay (see Materials and methods). By exploiting the within-participant variability in rotation onset time (SD ranged from 0.046 to 0.055 s across participants), we examined whether participants improved performance by taking more time to accumulate and process tactile information. For each participant, we ran an ANCOVA with absolute alignment error as the dependent variable and rotation onset time as a continuous predictor and edge length and initial dial orientation as categorical predictors. None of the participants showed a significant relationship between the absolute alignment error and the rotation onset time ($0.01 < F_{1,399} < 3.86$; $0.05 < P < 0.93$, uncorrected for multiple comparisons). In a corresponding analysis, we found that none of the participants showed a significant relationship between rotation duration and absolute alignment error ($0.07 < F_{1,399} < 3.41$; $0.07 < P_{uncorrected} < 0.79$). Likewise, we found no reliable effect of the whole trial duration (that is, the entire time from touch to the end of the rotation) on absolute alignment error ($0.01 < F_{1,399} < 6.17$; $0.01 < P_{uncorrected} < 0.92$).

We also tested if sub-movements during dial rotation improved alignment accuracy (*Figure 1*, *Figure 3—figure supplement 1a–d*). We reasoned that tactile processing of edge orientation might continue while the first movement was executed, which could improve the programming of subsequent movements (the second sub-movement, on average, commenced 0.22 ± 0.02 s after rotation onset). We found sub-movements in 44% of all trials. The frequency distribution of trials with and without sub-movements was similar for all edge lengths (*Figure 3—figure supplement 1c*) and trials with sub-movements were present in all participants (*Figure 3—figure supplement 1d*). Repeated measures ANOVAs with edge length, initial dial orientation, and presence of sub-movements as factors failed, however, to indicate a significant effect of sub-movements on the absolute alignment error (*Figure 3—figure supplement 1e*) or on the proportion of rotations in the correct direction (*Figure 3—figure supplement 1f*).

Taken together, these results suggest that study participants generated the dial rotation action based on tactile information extracted and processed essentially within ~200 ms of initial contact.

## Touch is nearly as good as vision

To benchmark pointer alignment accuracy based on touch, we had the same participants as in the main experiment to perform a visual version of the pointer alignment task. The experiment was identical to the main experiment with two exceptions. First, the shutter glasses opened at the onset of the auditory signal that instructed the participant to execute the task and remained open until the dial was contacted, which in practice implied that the participants could view the pointer position and the target location for 0.64 ± 0.20 s before rotation onset. Second, only the raised dot was used. Hence, in contrast to the main experiment where participants obtained information about the initial dial orientation solely by touching the dial, in the visual pointer-alignment task they obtained this information solely by seeing the pointer before touching the dial. We compared performance in the visually guided trials with that in the infinite edge condition from the main experiment, which yielded the best accuracy based on tactile information.

Alignment performance was marginally better in the visual than in the tactile condition (*Figure 4a,b*). The smaller absolute alignment in the visual condition (2.1 ± 0.5°) than in the tactile condition (2.9 ± 0.5°; $F_{1,9}$ = 12.9, p=0.006) mainly stemmed from smaller errors in the visual trials with initial dial orientations closest to the 0° target (*Figure 4—figure supplement 1a*). The rotation onset time in the visual condition (0.07 ± 0.04 s) was shorter than in the in tactile condition (0.20 ± 0.02 s; $F_{1,9}$ = 201.5, p<$10^{-6}$) (*Figure 4—figure supplement 1b*), presumably because participants could program the movement based on visual information obtained before touching the dial. The time from touch until contact force reached its plateau was modestly shorter in the visual condition (0.25 ± 0.10 s as compared to 0.31 ± 0.10 s; $F_{1,9}$ = 16.8, p=0.003; *Figure 4—figure supplement 1c*). Nevertheless, for all participants in the visual condition the onset of the rotation occurred during contact force increase. In fact, the sensory condition did not significantly influence the contact force at rotation onset or the plateau force (*Figure 4—figure supplement 1d,e*), and there were no statistically significant effects related to initial dial orientation on these timing and contact force parameters.

The kinematic structure of the rotation movement was remarkably similar in the visual and tactile pointer-alignment trials. First, we found no statistically significant effect of sensory condition on rotation duration (*Figure 4—figure supplement 1f*). Second, as for the tactile condition, in the visual condition none of the participants showed a significant effect of rotation onset time (0.011 < $F_{1,100}$<3.72; 0.06 < $P_{uncorrected}$ < 0.92) or on the duration of the pointer rotation (0.002 < $F_{1,100}$<2.47; 0.12 < $P_{uncorrected}$ < 0.96) on alignment error. Third, the frequency distribution of sub-movements did not significantly differ between the visual and tactile conditions (*Figure 4—figure supplement 1g,h*).

Taken together, the comparison of the tactile and the visual pointer-alignment trials revealed similar dial orientation accuracy and kinematics.

## A simple model of edge orientation processing

The prevailing model supporting tactile acuity at the level we report above suggests that the spatial details of touched objects are resolved based on the relative discharge rates of first-order tactile neurons having partly overlapping receptive fields and Gaussian-like sensitivity profiles

(*Friedman et al., 2002*; *Dodson et al., 1998*; *Wheat et al., 1995*; *Saal et al., 2017*; *Loomis and Collins, 1978*; *Khalsa et al., 1998*). Here, we propose an alternative explanation based on a generally overlooked feature of the peripheral apparatus – namely, that first order tactile neurons branch in the glabrous skin of the hand and innervate many spatially segregated mechanoreceptive transduction sites (*Cauna, 1956*; *Cauna, 1959*; *Nolano et al., 2003*). This arrangement yields first-order tactile neurons with heterogeneous cutaneous receptive fields that include many highly sensitive zones or 'subfields', apparently randomly distributed within a circular or elliptical area typically covering five to ten papillary ridges (*Johansson, 1978*; *Phillips et al., 1992*; *Pruszynski and Johansson, 2014*). At the neuronal population level, the high degree of receptive field overlap in the fingertips implies that first-order tactile neuron subfields are highly intermingled (*Figure 5a*). Thus, for edge orientation processing, an edge contacting the skin at a certain location and orientation will primarily excite that subset of the neurons whose subfields spatially coincide with the edge, while a different subset of neurons will be primarily excited for a slightly different edge orientation (*Figure 5b*). Such a coincidence code is attractive because it could enable the requisite edge orientation acuity and would operate at a speed suitable for the control of manipulation (*Johansson and Birznieks, 2004*).

We modelled a virtual patch of skin with known biological constraints to show how, under a simple coincidence-coding scheme (*Johansson and Birznieks, 2004*; *Stanley, 2013*; *Gire et al., 2013*; *Panzeri et al., 2001*), the presence of heterogeneous receptive fields with many subfields influences edge orientation resolution as a function of edge length (see Materials and methods and *Figure 5— figure supplement 1*). Briefly, the virtual patch was innervated by synthetic units (i.e., first-order tactile neurons) with innervation density (*Johansson and Vallbo, 1979*) and receptive field size (*Vallbo and Johansson, 1984*) based on the known human physiology. Each unit's receptive field was actually composed of receptor elements (i.e. mechanoreceptive transduction sites), the number, size and location of which was parameterized. We simulated the population response to edges that varied in length and orientation. Each unit in the population could be in two discrete states: active if the stimulus intersected any of its subfields or inactive otherwise. We deemed that the population response reliably differentiated between edge orientations when 5% of the relevant units changed their state between two orientations (see Materials and methods).

We compared two versions of the model. One where units had unique subfields by virtue of being connected to a random (2–64) number of receptors each 250 µm in diameter and placed randomly in the units nominally circular receptive field. And, as a comparison, another model where all units had receptive fields with uniform sensitivity by virtue of being connected to one receptor element whose receptive zone corresponded to the unit's receptive field boundary. *Figure 5c* summarizes three key insights of our modelling effort. First, the model with subfields performed at levels slightly better than our human participants – showing discrimination thresholds 1.3° for the infinite length edge to 13.1° for the 1 mm long edge. Second, the model with subfields always outperformed the model with a uniform receptive field. Third, and perhaps most interestingly, the performance gap between the two models grew for shorter edges (infinite edge difference = 6.4°; 1 mm = 28.4°), suggesting that heterogeneous receptive fields are particularly beneficial for demanding tasks that utilize tactile information approaching the limits of the system's spatial resolution.

## Discussion

Our study provides the first quantitative account of fine tactile spatial processing during object manipulation. Our findings reveal exquisite sensitivity to edge orientation. For edges spanning the entire contact area of the fingertip, accuracy in our tactile pointer-alignment task was on par with that when the participants used vision to orient the pointer (*Figure 4*). Performance was impressive even with much shorter edges. Interestingly, the threshold edge-length for 75% correct rotation direction was ~2 mm (*Figure 2d*), which corresponds to the dimensions of the smallest of manageable objects in everyday tasks. For example, the dimensions of jewellery clasps or buttons designed to be as small as possible for aesthetic reasons, rarely have edge lengths that go below ~2 mm.

### Action versus perception

Tactile edge orientation acuity has previously been examined in perceptual discrimination and identification tasks. The reported orientation acuity is 10–20° for edges that span a large portion of the fingertip (*Lechelt, 1992*; *Bensmaia et al., 2008*; *Peters et al., 2015*) and around 90° for a 2 mm long

edge (*Peters et al., 2015*). This sensitivity to edge orientation is substantially worse than that in our tactile pointer-alignment task (3 and 11° for the infinite and 2 mm edge lengths, respectively). A number of differences between our pointer alignment task and these previous perceptual tasks may contribute to the marked difference in reported acuity. In the following, we consider these differences—which vary across perceptual tasks—while keeping in mind that multiple factors likely contribute and our present results cannot reveal which factors are most influential.

Whereas participants in our pointer-alignment task actively moved their finger to contact the object, in all previous tasks examining edge orientation perception, the edge stimuli were externally applied to an immobilized finger. It is possible that subtle differences in contact dynamics contribute to increased acuity during active object manipulation. Moreover, although our results indicate that participants rapidly and accurately extract information about orientation prior to object rotation, some additional information about orientation may be available during the rotation, which could be used to adjust ongoing motor commands generating the rotation (see further below).

In the perceptual tasks previously used to assess tactile edge orientation sensitivity, participants reported orientation after the stimuli had been removed from the fingertip. Such reports must be based on a memorized representation of the tactile stimuli, which decays over time (*Sinclair and Burton, 1996*; *Gallace and Spence, 2009*). For example, the two-interval forced choice procedures that have been used require that the participant holds information about the stimuli in memory before a judgment is made based on a comparison of the sequentially presented stimuli (*Lechelt, 1992*; *Bensmaia et al., 2008*; *Peters et al., 2015*). Memory decay was reduced in our task because the response (i.e., dial rotation) was initiated very soon after the stimulus is contacted and while it remains in contact with the fingertip. Of course, our pointer alignment task requires that the participant remember the target orientation (i.e., straight ahead). However, this is constant across trials and the participant receives feedback at the end of each trial about the correct orientation. Moreover, the orientation of the raised edge contacted by the fingertip when held at its home position between trials may also provide information about the target orientation, helping the participants maintain an accurate spatial representation of the task environment. A match-to-sample task is another approach used in perceptual studies, where the participant is asked to identify which one of a set of lines, presented on a visual display in front of them, matches the orientation of an edge presented to, and then removed from, their fingertip (*Bensmaia et al., 2008*). In addition to memory demands, this task requires a transformation from the horizontal plane, in which the edge is presented, to the vertical plane in which the set of lines are displayed. Our task did not require such a transformation because the object was in the same plane when rotated as when it was first contacted.

Another factor that might contribute to differences in tactile edge orientation sensitivity in different situations is that the central processing of sensory information depends on task and context (*Engel et al., 2001*; *Crapse and Sommer, 2008*; *Schroeder et al., 2010*; *Gazzaley and Nobre, 2012*; *Zagha et al., 2013*; *Manita et al., 2015*). For example, it is well established that visual information can be processed differently in motor and perceptual tasks (*Bridgeman, 2000*; *Kravitz et al., 2011*; *Milner and Goodale, 2006*; *Rossetti and Pisella, 2002*; *Weiskrantz, 1996*) and that such difference partially reflect the engagement of different cortical processing pathways (*Milner and Goodale, 2008*; *Ungerleider and Haxby, 1994*). The dorsal visual pathway, supporting action, is thought to compute real-time motor commands based on directly viewed object metrics, and the ventral visual pathway, supporting perception, is thought to extract lasting and detailed information about objects via memory associations and recognition. For the somatosensory system, motor-related signals at different levels of the sensorimotor system can affect virtually all levels of the tactile processing pathway (*Zagha et al., 2013*; *Manita et al., 2015*; *Canedo, 1997*; *Fanselow and Nicolelis, 1999*; *Lee et al., 2008*; *Seki and Fetz, 2012*; *Adams et al., 2013*) and incoming tactile information can be predictively filtered during action, making it possible for the sensorimotor system to attenuate irrelevant inputs and highlight task-critical information (*Chapman, 1994*; *Bays et al., 2005*; *Blakemore et al., 1999*). Since our task can be considered a tool-use task, it is also interesting to note that tool use is associated with dynamically remapping of space represented by central multisensory, predominantly tactile neurons (*Maravita and Iriki, 2004*; *Brozzoli et al., 2012*). However, we are not aware of any studies that have directly compared the neural processing of exactly the same macrogeometric tactile information depending on whether used in fine dexterity tasks or in perceptual judgements.

## Processing speed

Our findings also reveal the speed with which the motor system can process and use macrogeometric tactile information; something not addressed in previous perceptual studies. The time from touch to rotation onset in the tactile pointer-alignment task was ~200 ms. In this time, participants established contact with the dial, acquired and processed edge orientation information, and programmed and initiated the rotation movement. Since the rotation movement could be programmed before touch in the visual trials, it seems reasonable to suggest that the added time between touch and rotation onset in the tactile trials (~130 ms) represents the time actually required to extract and process tactile edge orientation information. Such fast acquisition and use of tactile information is in agreement with the automaticity by which tactile signals are used in other aspects of object manipulation, including mechanisms supporting grasp stability (*Johansson and Flanagan, 2009*) and target-directed reaching guided by touch (*Pruszynski et al., 2016*). Likewise, as with other action patterns rapidly triggered by tactile events during unfolding manipulation (*Pruszynski et al., 2016*; *Johansson and Westling, 1987*), we found no effect of the fidelity of the sensory information (i.e. edge length) on the latency of the triggered action (i.e. the start of rotation). This contrasts with typical results of perceptual studies where the reaction time measures typically increase when the credibility of the sensory information decreases (*Pins and Bonnet, 1996*).

We found that participants did not improve their performance by taking more time to process tactile information during the trial duration or by making sub-movements during the rotation. These results suggest that the important tactile information used in our task was acquired very soon after the edge was initially touched (*Johansson and Birznieks, 2004*) and thus signalled by the dynamic responses in first order tactile neurons when the edge deformed the skin during the contact force increase. Indeed, tactile afferent information available later during the rotation would have been restricted largely to gradually fading responses in some of the slowly adapting tactile neurons. Interestingly, the dynamic response of first order tactile neurons also seems highly informative for the perception of edge orientation since the duration of stimulation seems to marginally influence performance in psychophysical tests – indenting the fingertip with a 10 mm long bar for 400 ms compared to 100 ms only slightly improved average orientation identification threshold (from 26.6 to 23.4°) (*Bensmaia et al., 2008*).

## Tactile versus visual acuity

Our comparison of the tactile and the visual pointer-alignment trials revealed similar dial orientation accuracy. We justify our comparison by the fact that both tasks primarily gauged the accuracy of movement planning before rotation onset, where touch specified the initial orientation of the dial in the tactile task and vision in the visual task. Since there was no visual feedback during the rotational movement in any of the tasks, in both tasks the information available for possible online control of the pointer's state (position, velocity) was restricted to proprioceptive and/or efference information about the movement of the hand (*Desmurget and Grafton, 2000*). The fact that the edge was touched during the rotation in the tactile trial did not mean that tactile afferent signals from the fingertip conveyed information about the pointer's state during the rotation. Tactile signals related to the orientation of the edge would provide information about the pointer's orientation relative to the fingertip rather than information about the pointer's orientation in external space. Because of its very low rotational friction and moment of inertia (see Materials and methods), the dial offered negligible resistance to rotation, limiting skin deformation changes related to rotation of the dial (i.e., no rotational slips and virtually no twist forces impeding the movement occurred in the digit-dial interface during the rotation). However, we cannot fully exclude that signals mainly in slowly adaptive tactile neurons could have helped in the tactile task by facilitating possible proprioceptively based online control by gradually improving the assessment of the orientation of the edge relative to the fingertip during the ongoing movement.

## Neural mechanisms

The actual sensitivity of the tactile apparatus to edge orientation must be better than indicated by our experiment, since our approach, though naturalistic, introduces several sources of information loss in this regard. This would include noise related to arm-hand coordination and postural actions in our standing participants, as well as information loss associated with memory decay. Yet, the edge

orientation sensitivity as revealed in our tactile pointer-alignment task substantially exceeds that predicted by the Shannon-Nyquist sampling theorem if assuming a pixel-like mosaic of tactile innervation determined by the density of relevant first-order tactile neurons in the human fingertips. For example, with the 4 mm edge the average alignment error (5.9°) corresponds to a position change of just 0.21 mm at the end of the edge if rotated around its center, which is very small in relation to the ~1 mm average spacing between receptive field centers in human fingertips (*Johansson and Vallbo, 1979*) (*Figure 2c*). The ability of humans to perform spatial discrimination finer than that predicted by the average spacing between receptive field centers, termed hyperacuity, has been examined extensively in vision (*Westheimer, 2012*), but has also been reported for touch (*Dodson et al., 1998*; *Wheat et al., 1995*; *Loomis, 1979*). The currently accepted model supporting tactile hyperacuity, built largely on neural recordings in monkeys, centers on the assumption that first-order tactile neurons have simple Gaussian-like sensitivity profiles; at the population level, this model proposes that spatial details are resolved by comparing the discharge rates of neurons – generally estimated over hundreds of milliseconds (*Khalsa et al., 1998*) – with neighbouring receptive fields via an unknown neural interpolation scheme (*Friedman et al., 2002*; *Dodson et al., 1998*; *Wheat et al., 1995*; *Saal et al., 2017*; *Loomis and Collins, 1978*).

We propose an alternative explanation for tactile hyperacuity. We are motivated by the fact that first order tactile neurons branch in the skin and innervate many spatially segregated mechanoreceptive transduction sites (*Cauna, 1956*; *Cauna, 1959*; *Nolano et al., 2003*; *Lesniak et al., 2014*; *Paré et al., 2002*; *Lindblom and Tapper, 1966*; *Looft, 1986*; *Brown and Iggo, 1967*; *Goldfinger, 1990*; *Vallbo et al., 1995*), a feature of the peripheral apparatus not incorporated into previous models of tactile acuity. For the human fingertips, this arrangement yields first-order tactile neurons with heterogeneous cutaneous receptive fields that include many highly sensitive zones distributed within a circular or elliptical area typically covering five to ten papillary ridges (*Johansson, 1978*; *Phillips et al., 1992*; *Pruszynski and Johansson, 2014*). Critically, at the population level, these receptive fields are highly intermingled (*Figure 5a*) meaning an edge contacting the skin at a certain location and orientation will excite one subset of the neurons while contacting the skin at a different location or orientation will excite a slightly different subset of neurons (*Figure 5b*). Under our proposed scheme, the degree to which different edge orientations synchronously engage different subsets of neurons determines edge orientation resolution, which would be higher than predicted by the center-to-center spacing of the receptive fields because the average spacing between subfields is substantially less than the average spacing between receptive field centers. This coincidence code is attractive because established neural mechanisms for central sensory processing provide rich possibilities for moment-to-moment segregation and representation of edge orientation (and other spatial features) at a speed suitable for rapid integration in the control of manipulation. That is, the massive divergence and convergence of first-order neurons in the periphery onto second and higher order neurons in the central nervous system (*Jones, 2000*), together with these neurons functioning as efficient coincidence detectors (*König et al., 1996*; *Usrey, 2002*), could allow fast feedforward processing of spatially correlated spiking activity in ensembles of first-order neurons (*Pruszynski and Johansson, 2014*; *Johansson and Birznieks, 2004*; *Johansson and Flanagan, 2009*; *Jörntell et al., 2014*). Of course, our present work cannot establish that this specific code is used by the central nervous system but recent work demonstrating the ability to record from large populations of second-order tactile neurons in the brainstem of awake primates offers the means to directly test this idea (*Suresh et al., 2017*).

A fundamental question is why the nervous system evolved to sample tactile inputs via neurons that have small and heterogeneous receptive fields. We believe that the convergence of inputs from multiple mechanoreceptive transduction sites on individual first-order neurons (yielding subfields) represents an optimal scheme for preserving behaviourally relevant spatial tactile information given the relatively tight space constraints for neurons in the peripheral nerve (axons) and dorsal root ganglion (cell bodies) as compared to mechanoreceptors in the skin (*Zhao et al., 2017*). For example, recent work from the field of compressed sensing shows that randomly sampling a sparse input signal often allows it to be fully reconstructed with fewer measurements than predicted by the Shannon-Nyquist theorem (*Candes and Wakin, 2008*; *Candès et al., 2006*), suggesting that heterogeneous connections in the tactile periphery may help overcome sensory processing bottlenecks related to pathway convergence (*Candes and Wakin, 2008*; *Candès et al., 2006*).

# Materials and methods

## Participants

Twenty healthy people (nine female, age range: 20–38) volunteered for these experiments. Participants provided written informed consent in accordance with the Declaration of Helsinki. The ethics committee at Umea University approved the study.

## General procedure

Study participants stood at a table (90 cm high) and rested their left hand on the tabletop. The tip of their right index finger was held at a home position located above a horizontally oriented dial located on the tabletop (*Figure 1a–1*). Participants were instructed to move their right index finger down from the home position to contact the dial at its center of rotation (*Figures 1a–2*) and rotate the dial such that the pointer, extending from the horizontally oriented contact surface, pointed at the center position of the dial, labeled 0° (*Figures 1a–3*), which corresponded to orienting the pointer straight ahead. Thus, correctly orienting the dial required rotating the dial in the direction opposite the initial orientation. Although participants were not constrained and we did not measure their joint angles, they appeared to rotate their finger around the rotational axis of the dial by abducting (or adducting) their shoulder joint and adducting (or abducting) their wrist joint. The task was considered completed when the pointer was positioned within ±2° of the 0° target (*Figures 1a–4*, *Video 1*). A black clip attached to the dial indicated this target zone. Oriented in the direction of the pointer, a 1 mm thick raised edge on the otherwise flat contact surface of the dial provided tactile information about the initial orientation. The length of this edge and the initial orientation of the dial when initially contacted constituted experimental variables. Participants wore shutter glasses, which could prevent the participant from seeing the apparatus before and during the rotation.

## Apparatus

The pointer (11.5 cm long) was attached to the periphery of a horizontally oriented exchangeable circular contact surface (diameter = 44 mm). The center of the contact surface was mounted on a vertical shaft of a practically frictionless potentiometer (Model 3486, Bourns Inc., Toronto, Canada) that measured the orientation of the dial (resolution <0.1°) (*Figure 1b*). Both the pointer and the contact surface were made of plastic and the entire assembly had a very low moment of inertia (337 g*cm [*Chemnitz et al., 2013*]). Due to the very low rotational friction and moment of inertia of the dial, the device exhibited virtually no mechanical resistance to rotation. A force transducer (FT-Nano 17, Assurance Technologies, Garner, NC, USA) mounted in series with the potentiometer measured the normal force applied to the contact surface. A model aircraft servo with a fork-like assembly attached to the rotation axis could set the pointer to any position within ±38° relative to the target position (i.e. straight ahead, 0°). When the servo had moved the dial to the set orientation, it returned to a home position so that it did not affect the range of pointer rotation, which was ±38°. All servo actions took place between trials and, to avoid auditory cues from the motor about the initial dial orientation, the servo was programmed to always carry out a similar pattern of movements prior to each trial. Shutter glasses (PLATO, Translucent Tech., Toronto, Canada) occluded the participant's vision at specific times during the pointer-alignment trials. A loudspeaker provided auditory commands and trial feedback.

The raised edge of the contact surface was 1 mm high and 1 mm wide. It had a hemi-cylindrical top in cross section (radius = 0.5 mm) and curved ends (radius = 0.5 mm) (*Figure 1d*). The length of the straight portion of the edge was varied between conditions and could be 0, 1, 2, 4, 8 or 44 mm (*Figure 1c*). Since the 44 mm edge spanned the entire area of contact with the fingertip, we refer the length of this edge as being infinite. The 0 mm edge was actually a 1 mm diameter raised dot with hemispherical top. All edges > 0 mm were aligned with the long-axis of the pointer and were centered on its rotational axis, thus providing veridical information about the orientation of the pointer.

When the index finger was at its home position, it rested on the upper surface of a horizontally oriented rectangular plate (20 × 32 mm) mounted above the distal segment of the circular contact surface (*Figure 1b*). A raised edge, centered on the plate and spanning its entire length, was pointing towards the target position (i.e., 0°). The cross section profile of this edge was the same as the

edges on circular contact surface. The function of this edge was to offer the participants a tactile reference for the finger's home position.

## Main experiment

### Tactile pointer-alignment

Ten study participants volunteered in this main experiment (five female). In periods between trials, with the shutter glasses closed, the pointer was rotated to one of six angular positions relative to the target position (−30, −20, −10, 10, 20 and 30°). Therefore, reaching the target position (0°) from these initial dial orientations, required rotation of the dial clockwise by 30, 20, 10° and counter-clockwise by 10, 20 and 30°, respectively.

An auditory signal consisting of three short beeps (1.2 kHz, 300 ms), instructed the participant to perform a trial, which entailed moving their finger from the home position to the contact surface and turning the pointer to the target position. Participants were free to choose the speed with which to move their finger and rotate the pointer, but were told to turn the dial when contacted.

The shutter glasses opened when the rotation movement ended, defined as the time when the speed of the rotation fell below 10°/s for a period ≥200 ms. The rotation speed, computed online by numerical differentiation, was filtered by a first-order low pass filter with a 10 ms time constant (cut-off frequency = 16 Hz). If a movement ended outside the ±2° target zone, the participant made final adjustments under visual guidance. When the pointer had been kept within the target zone for 300 ms, the shutter glasses closed again and the participant received auditory feedback indicating goal completion (beep @ 240 Hz for 50 ms). If the initial movement ended within the ±2° target zone, the shutter glasses opened for 300 ms and when the shutters closed again, the participant received auditory feedback indicating goal completion (beep @ 240 Hz during 50 ms). Thus, in either case, the participant obtained visual feedback about the outcome of the rotation.

The auditory feedback about goal completion indicated to the participant to return their finger to the home position. During this inter-trial period, the shutter glasses were closed and the servo rotated the dial to the initial dial orientation of the forthcoming trial. The servo started 0.8 s after the contact with the dial was broken (assessed on-line based on the force transducer signal) and operated for 1.8 s irrespective of the programmed dial orientation.

To engage participants and encourage good performance, after each block they received verbal feedback on the number of trials in which the rotation ended within the target zone. Furthermore, to keep the participants alert and to maintain a good pace in the experiment, the rotation had to be initiated less than 350 ms after the contact surface was touched. In trials where participants did not meet this timing requirement (<10%), they received auditory feedback and the trial was aborted. Aborted trials were re-inserted at a randomly selected point in the experiment. In this on-line control of the trial progression the time of touch and onset of rotation were defined by the time the normal force exceeded 0.2 N and the time rotation speed exceeded 10°/s, respectively.

In the main experiment, each participant performed 648 pointer-alignment trials (six edge lengths × six initial orientations × 18 repeats), which were broken down into blocks of trials where the edge length was held constant. For each edge length, participants performed three consecutive blocks of 36 trials per block (six initial orientations × six repeats). Within each block, the various initial orientations were randomly interleaved preventing the participants from predicting the direction and magnitude of the rotation required to reach the target. The blocks with the various edge lengths were presented in the following order for all participants: Infinite, 8, 4, 2, 1 and 0 mm length. To familiarize subjects with the task, the participants ran one practicing block of pointer-alignment trials with the infinite edge prior to beginning the main experiment (*Figure 1—figure supplement 1*). Participants could rest between blocks as desired.

### Visual pointer-alignment

For comparison with the tactile pointer-alignment task, we also studied the performance of the same ten individuals who participated in the main experiment when they could see the dial, including the position of the pointer, and the target position before initiating the rotation. The trials were identical to the trials of the main experiment with two exceptions. First, the shutter glasses opened at the beginning of the auditory cue telling the participant to perform a trial and were open until the contact surface was touched. Second, only the raised dot was used (edge length = 0 mm), meaning

that 108 visual pointer-alignment trials were performed (six initial dial orientations × 6 repeats × three blocks). As with the tactile pointer-alignment trials, participants were familiarized with the visual trials by performing one block of trials under the visual condition before first formal block was executed. The order by which the blocks of tactile and visual pointer-alignment trials were presented was counterbalanced across participants.

### Follow-up experiment

In our main experiment, the dial was initially oriented at one of six orientations. Thus, it is possible that participants may have learned six rote responses and then selected one of these responses based on coarse discrimination among the six edge initial orientations (10° apart). Although the results of the main experiment indicate that this is unlikely (see Results), we carried out a follow-up experiment with 50 initial dial orientations to rule out this possibility.

Ten additional participants performed the same tactile pointer alignment task used in the main experiment with the following differences. Only two edges were used: the infinite edge and the raised dot (0 mm edge); the inclusion of the raised dot allowed us to verify that the experiment did not include cues about the dial orientation in addition to those provided by the edge when present. For each edge, two consecutive blocks of trials were run, including 100 trials in total. The initial orientation of the edge was randomized, without replacement, between −32 to −8° and +8 to +32° in 1° increments (0° is straight ahead), resulting in 50 different initial orientations. As in the main experiment, the participants were familiarized with the task by performing one block of 50 trials with the infinite edge before the first formal block was executed. This experiment was carried out in conjunction with an experiment on perceptual edge orientation acuity not presented here.

### Data analysis

The signals representing the orientation of the dial, the orientation of the 'reporting line', and the normal force applied to the contact surface were digitized and stored with 16-bit resolution at a rate of 1000 Hz (S/C Zoom, Umeå, Sweden). Using parameters that we defined during a preliminary analysis of the data, we extracted the following variables for data analysis.

The time of initial contact with the dial (initial touch) represented the event when the right index finger first contacted the contact surface. This was measured as the first instance the normal contact force exceeded 0.01 N of the median force value during a 500 ms period ending immediately before the time of the go signal. To prevent triggering on possible noise in the force signal occurring when the participant moved the finger from the home position, we first searched for a contact force exceeding 0.2 N and then searched backwards to the criterion force level.

The duration of contact force increase in the pointer-alignment trials was the period between time of touch of and the time when the contact force reached a plateau-like state. To calculate the latter time, we first calculated the force rate (i.e. derivative of force) with cut-off frequency of 8.7 Hz. We searched forward for the maximum local peak of force rate increase during the period 50–350 ms after touch. We then searched further forward and defined the end of force increase as the instance that the force rate first decreased below 10% of the maximum local peak force rate. At this instant, we also recorded the plateau contact force. The selected time window for peak detection avoided capturing the end of a transient, generally small, impact force that could occur when the finger initially touched the dial. It also avoided triggering on transient contact force changes that occasionally occurred late during the trials.

The rotation velocity of the dial and of the reporting line was calculated by symmetric numerical time differentiation of the dial orientation signals (±1 samples) after being low-pass filtered with a cut-off frequency of 17 Hz (symmetrical triangular filter). Inspection of the velocity profiles during dial rotation revealed that the rotation could possess sub-movements, i.e., it could contain multiple distinct velocity peaks (see *Figure 3—figure supplement 1a–b*). We defined peaks (positive and negative) in the velocity profile by searching for zero-crossings (with negative slope) in the first time differential of the dial rotation speed computed as the absolute value of the rotation velocity and low-pass filtered with cut-off frequency of 8.7 Hz. For each defined peak, we recorded its time and the pointer velocity. By identifying minima in a symmetrical high-pass filtered version of the pointer speed signal (triangular filter, cut-off frequency of 2.1 Hz) we could accurately estimate the time of rotation onset, durations of sub-movements if present, and the time of the end of the rotation. That

is, the rotation onset was measured as the point when the high-pass filtered pointer speed had its first minimum found by searching backwards from the time of the first peak in the time differentiated pointer speed signal. At this time, we also recorded the contact force. In pointer-alignment trials that contained sub-movements, subsequent minima defined times that separated successive sub-movement and the last minimum encountered >200 ms before the time that the shutter opened defined the end of the rotation movement. Likewise, in trials without sub-movements (single velocity peak) the second (and last) minimum defined the time of the end of the rotation movement.

The duration of dial rotation was the time between of rotation onset and end of rotation and the resultant dial orientation, providing the alignment error in the pointer-alignment tasks, was defined as the orientation at the time of rotation ended. The displacement of the pointer was calculated as the difference between resultant dial orientation and the initial orientation referenced to the direction towards the target, that is positive and negative values indicated rotation towards and from the target, respectively. Peak contact force was the maximum contact force recorded during the period of contact.

## Statistical analysis

Effects of the experimental factors on behavioral variables were assessed using repeated-measures analyses of variance (ANOVAs). Unless otherwise indicated, edge length and initial dial orientation constituted the categorical predictors (factors) in the analysis pertaining to the main experiment whereas sensory condition (tactile, visual) and initial orientation were categorical predictors in comparisons between the tactile and visual pointer-alignment tasks. In analyses of covariance (ANCOVAs) performed at the level of individual participants (see Results), we used Holm-Bonferroni correction for multiple comparisons. In statistical analyses that involved the absolute alignment error as a dependent variable, the data were logarithmically transformed to approach a normal distribution. Data were Fisher and arcsine transformed when performing parametric statistics on correlation coefficients and proportions, respectively. Throughout, we defined a statistically significant outcome if $p < 0.01$ and for post-hoc comparisons, we used the Tukey HSD test. Unless otherwise stated, reported point estimates based on sample data refer to mean $\pm 1$ standard deviation of participant's medians computed across all edge orientations and relevant edges.

## Model

We modelled a virtual patch of skin ($2 \times 2$ cm) constrained by known biological features of the human tactile periphery (for visual description, see *Figure 5—figure supplement 1*). The patch was connected to synthetic units meant to represent first-order tactile neurons. The center of each unit's receptive field was randomly placed on the patch. Units were placed until the average distance between the center of each receptive field and the center of its six nearest neighbours was, on average, ~1 mm as previously described (*Johansson and Vallbo, 1979*; *Johansson and Vallbo, 1980*). Each unit had a nominally circular receptive field drawn from a log normal distribution as previously described (*Vallbo and Johansson, 1984*) (in $\log_{10}$ units: mean = 1, SD = 0.45). The receptive field was composed of receptor elements (E) meant to represent a neuron's mechanoreceptive transduction sites. Although a unit (U) could have many transduction sites, its output could be in only two discrete activation states (A): active when the stimulus intersects with any of its receptor elements or inactive when the stimulus does not intersect with any of its receptor elements (*Equation 1*).

$$A(U_i) = 1 \text{ if } A(E_{i,1}) \ \| \ A(E_{i,2}) \ ... \ \| \ A(E_{i,n}) = 1, \tag{1}$$

where i represents the synthetic units and n represents each synthetic unit's receptor elements. (*Equation 1*).

We compared two versions of the model that differed at the level of the receptor elements. The main version (with subfields) had units with receptive fields composed of many receptor elements ($E_{i,n}$; *Equation 2*). The presence of many receptor elements was meant to represent the fact that first-order tactile neurons branch and innervate many mechanoreceptive end organs, and have complex receptive fields with many highly-sensitive zones (or subfields) (*Johansson, 1978*; *Phillips et al., 1992*; *Pruszynski and Johansson, 2014*). In this version of the model, the number, size and location of receptor elements were parameters chosen as follows. The location of the elements was randomized except for the first two elements, which were placed opposite to one another on the receptive

field boundary. The diameter of the circular receptor elements was fixed to 250 microns (that is, they were considered active if the Euclidian distance (*d*) between the stimulus and the receptor element center was <125 microns). The number of receptor elements was randomized between 2 and 64 (uniform distribution). Such complex receptive fields correspond to the known sensitivity profiles of human first order tactile neurons (*Figure 5a*). The second version of the model (without subfields) had units with a single receptor element ($E_{i,1}$; *Equation 3*). In this version of the model, the size and location of each unit's receptor element corresponded precisely to its receptive field boundary thus the activation of each synthetic unit was equivalent to the activation of its only receptor element. Such plate-like receptive fields, which consider only the boundary of first-order tactile neuron receptive fields and ignore their internal topography rendering uniform sensitivity throughout the field, have previously been used to describe the sensitivity profile of first-order tactile neuron receptive fields (*Johansson and Vallbo, 1980*; *Gardner and Palmer, 1989*).

Hence, our main interest was testing how well the following two versions of the model could signal edge orientation as a function of edge length:

$$A(E_{i,n}) = 1 \text{ if } d(\text{edge}, E_{i,n}) < 125 \text{ microns} \tag{2}$$

$$A(E_{i,1}) = 1 \text{ if } d(\text{edge}, E_{i,1}) < \text{Radius of } U_i'\text{s RF} \tag{3}$$

We did this by generating the same virtual fingertip for both versions of the model. That is, runs were paired such that the receptive field sizes and locations, along with the location of stimulus, were identical for both versions of the model. Moreover, our stimuli for different edge lengths always rotated the edge about its center at the same location and in the same direction. At the beginning of each simulation, for each model, we determined which units were active at the initial edge placement, which we termed 0° (represented by activation vector across units, called $A_{@0°}$). We also determined the number of units that could be potentially activated by the edge – that is the number of neurons that could be contacted if the edge rotated completely about is center (called $N_c$) which served as a normalization factor across edge lengths. We then rotated the edge about its center in 0.5° increments and recalculated which units were active at each step. We deemed that the edge was discriminated at Δ° when 5% of the potentially active units changed their state from the initial stimulation that is, if the Hamming distance between $A_{0°}$ and $A_{0+Δ°}$ was $\geq 0.05*N_c$ (*Equation 3*). We repeated this process with 100 virtual fingertips for each model.

## Acknowledgements

We thank Anders Bäckström, Carola Hjältén and Per Utsi for their technical and logistical support.

## Additional information

### Funding

| Funder | Grant reference number | Author |
| --- | --- | --- |
| Canadian Institutes of Health Research | Foundation Grant 3531979 | J Andrew Pruszynski |
| Vetenskapsrådet | Project 22209 | J Andrew Pruszynski |
| Canadian Institutes of Health Research | OOGP 82837 | J Randall Flanagan |

The funders had no role in study design, data collection and interpretation, or the decision to submit the work for publication.

### Author contributions

J Andrew Pruszynski, Roland S Johansson, Conceptualization, Formal analysis, Funding acquisition, Investigation, Visualization, Methodology, Writing—original draft, Writing—review and editing; J Randall Flanagan, Conceptualization, Funding acquisition, Visualization, Methodology, Writing—original draft, Writing—review and editing

## Author ORCIDs

J Andrew Pruszynski [iD] http://orcid.org/0000-0003-0786-0081
Roland S Johansson [iD] https://orcid.org/0000-0003-3288-8326

## Ethics

Human subjects: Twenty healthy people volunteered for these experiments. All participants provided written informed consent in accordance with the Declaration of Helsinki. The ethics committee at Umea University approved the study (approval number 215/03, 03-167). Consent for publication has been obtained, in particular from the individual in Video 1.

## Decision letter and Author response

Decision letter https://doi.org/10.7554/eLife.31200.019
Author response https://doi.org/10.7554/eLife.31200.020

## Additional files

### Supplementary files

• Transparent reporting form
DOI: https://doi.org/10.7554/eLife.31200.017

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
