## [Decision Letter]

Thank you for submitting your article "Fast and accurate edge orientation processing during object manipulation" for consideration by *eLife*. Your article has been favorably evaluated by Sabine Kastner (Senior Editor) and three reviewers, one of whom, Timothy Verstynen (Reviewer #1), is a member of our Board of Reviewing Editors. The following individual involved in review of your submission has agreed to reveal their identity: Esther Gardner (Reviewer #3).

The reviewers have discussed the reviews with one another and the Reviewing Editor has drafted this decision to help you prepare a revised submission.

Summary:

The present study looks at the tactile system's capacity to process edge orientation information during an active object orientation task. Using a novel experimental paradigm, where participants briefly touched an edge and then rotate its underlying surface to align with the straight ahead direction in the extrinsic coordinate system the authors found that: 1) the performance in this active task was substantially better than in previously reported studies, with participants initiating the alignment movement ~200ms after touching, and presenting with an alignment error of less than 3 degrees; 2) performance in the task was approximately on par with that measured in a control condition (visual pointer alignment), where participants were allowed to use visual information to perform the task. Providing visual feedback during the task did not improve tactile estimation performance, leading the authors to conclude that the spatial sensitivity of the tactile system is comparable to that of vision in such active exploration tasks.

All three reviewers (including the Review Editor) thought that this is a strong study that extends our understanding of the sensitivity of tactile perception during active exploration in several novel ways. There was unanimous agreement that the design was innovative, the findings robust, and the implications were broad. However, all three reviewers converged on a common set of concerns about the current version of the paper that need to be revised and addressed.

Essential revisions:

1) The Model: The toy model presented at the end of the paper is somewhat fundamental for interpreting the findings in a physiological context. As reviewer #3 points out, it is a "simple" model, not a "toy" model, and has substantial experimental support from the authors' own work, and that of other investigators. It is both out of place being in the Discussion section and lacking clear methodological details. Please move the model into the core of the paper, with adequate descriptions in the Materials and methods on how it is implemented and a more complete description of the findings in the Results. The authors are encouraged to look at recent study from Ellen Lumpkin's lab on summated responses from Merkel cell afferents when stimulated together (Lesniak et al., *eLife*2014), and should also look at Suresh et al., 2016 as these findings may bolster the biological validity of the model being proposed.

2) Active vs. Passive: The main novel finding of the paper is the fast and accurate tactile capacity discovered in the new paradigm; however, this discrepancy with existing literature on tactile perception is likely due to task differences. Reviewers #1 and #2 pointed out that previous research in tactile discrimination and identification typically used passive paradigms, i.e., stimuli were applied on an immobilized finger pad. Even previous animal studies and their associated theoretical models utilized anesthetized monkeys (e.g., Friedman et al., 2002). The present study required the participant to actively touch the edge and then moved it without any constraints. The authors also noted that their task had continuous tactile inputs while previous studies briefly presented the stimuli to the fingertip before the perceptual report. Is this then somewhat an apples-and-oranges comparison? Active exploration is known to produce better perceptual performance in the other perceptual systems. Reviewer #3 pointed out that the increased sensitivity to edge orientation may reflect the pragmatic utilization of tactile signals by the motor system, rather than the cognitive utilization of matching tactile and visual edges as has been performed in previous psychophysical studies.

The authors should tone down the interpretation the implications of their findings in this regard. It is not unexpected that active exploration improves sensitivity, but what is striking is the degree to which it improves tactile sensitivity. The authors should revise how they link their results to this previous passive literature to acknowledge this difference and comment on the use of sensorimotor parameters rather than verbal responses in their perceptual measurements.

3) Measures of "Processing Speed": There was unanimous concern about the measures of processing speed reported here. If rotation onset time and time of contact force increase are measures of processing speed, wouldn't you expect them to get longer as available sensory information decreases (i.e., as edge length decreases)? Unlike perceptual reports, the time of starting to rotate the surface does not necessarily equate to the time that participants make up their mind about the edge direction. As this is an active task, participants could very likely receive more information during rotation since fast-adapting tactile endings (Meissner and Pacini endings) are better recruited during movements. This possibility is highlighted by the fact that 44% of all trials had sub-movements, which might relate to the fact that participants were adjusting their initial "judgment" based on incoming new feedback. Reviewer #2 pointed out that this possibility cannot be ruled out by ANCOVA results as reported. For this analysis, absolute alignment error was the dependent variable, rotation onset time was the covariate, and edge length and initial dial orientation were categorical variables. There was no significant correlation between absolute error and rotation onset time. Yes, onset time appears to not affect absolute error, but it is still possible that the performance was determined by perceptual inputs during the whole trial duration, no matter where the onset happened. This is perhaps one of the most critical problems in the paper. While none of the reviewers thought that this required collecting new data with a better measure of processing speed, the authors should acknowledge this limitation throughout the manuscript (not just as an addition to the Discussion) and temper their descriptions of processing speed per se.

4) Visual vs. Tactile: There were also several concerns about the clarity of the links between the tactile experiment and the visual experiment. All three reviewers felt that the motivation, description, and interpretation of this across modality comparison needed more detail. Reviewer #1 felt that the authors were downplaying an apparent consistent difference between the modalities. The authors try to say that performance was essentially identical across the tactile and visual conditions, however, Figure 4B and Figure 3—figure supplement 1B and C clearly show a consistent within-subject difference across conditions (with the mean trends masked by the between subject variance). So it appears that sensory modality does matter. Reviewer #2 pointed out that the tactile only and visual only experiments didn't just vary in their perceptual modality because participants received visual inputs until they touched the contact surface. Thus, the perceptual judgment was affected by memory decay of the visual representation of the dial. In contrast, in the main experiment, tactile feedback was continuously available for guiding the movement. Thus, while two experiments achieved similar performance, it does not necessarily mean that this non-effect is due to comparable unimodal performance as claimed by the authors. It is possibly due to the availability of sensory inputs too. The authors correctly pointed out that tactile stimuli were continuously available when they discussed the difference between their study and those early tactile perception studies. However, this discontinuous-versus-continuous distinction is also present in their comparison between the visual and tactile experiments. This is perhaps the most critical problem identified in the study. Had the authors made matching tactile and visual sensitivities be the critical focus of the study, these concerns would have been seen as a fundamental/critical flaw requiring new data. However, since the main focus is on describing the sensitivity of tactile perception during active exploration (and the comparison to visual sensitivity gives context), we suggest that the authors identify either analytical solutions to address these concerns and/or highlight these limitations and temper the interpretive links between tactile and visual modalities.

5) Task Description: The task description itself was found to be opaque, particularly for the broad/non-specialized audience of *eLife*. Reviewer #3 points out that this motor task is similar to adjusting the minute hand on a watch to the 12:00 position from either the 11:55 or 12:05 position. In the current task, the subject touches the test edge and uses the tactile sense to detect its orientation, and then turn the dial in the opposite direction [clockwise or counterclockwise] to the upright position. In other words, the subject must recognize the edge orientation, and make an equal and opposite hand movement to accomplish the task goals. The task instructions do not explicitly tell the subject to detect the initial angle of the edge, only that the dial needs to be turned to the upright (i.e., vertical) target position. Edge orientations studied were +/- 10, 20, or 30°; subjects were required to move the dial to within 2° of the vertical position. While the task is clearly described in the extended Materials and methods, the presentation needs a better descriptor than in the current version. The authors are encouraged to use the clock face metaphor or that of a compass needle rotated from NE or NW to due North in order to make the details more accessible to a broader audience. The Reviewing Editor notes that the authors may choose to use another technique to increase accessibility of the task descriptions.

[Editors' note: further revisions were requested prior to acceptance, as described below.]

Thank you for resubmitting your work entitled "Fast and accurate edge orientation processing during object manipulation" for further consideration at *eLife*. Your revised article has been favorably reevaluated by Sabine Kastner (Senior Editor) and a Reviewing Editor.

The manuscript has been improved but there are some remaining issues that need to be addressed before acceptance, as outlined below: In the evaluation of the revision (which was sent back to the original reviewers), there was a consensus amongst all reviewers that, while the manuscript was substantially improved, some of the original concerns remained.

The open issues from items in the original summary are as follows:

1) The Model: All reviewers thought that the relocation of the model into the Results and expanded description have helped better integrate this with the main psychophysical findings. However, there remain two points of concern with this model.

The first concern centers on clarity of the revised description. The authors note that cutaneous mechanoreceptors in the human hand branch at their distal endings, and that SA1 and RA1 afferents are capable of spike generation at any and all of these branches. They propose that the receptive field sensitivity reflects the stimulation pattern of any and all of these branches. This is a very intriguing idea, but the model is not sufficiently described in the report. The basic idea is that subjects can distinguish whether particular afferents are included or excluded if the parent axon fires a spike. They illustrate a plausible mechanism for distinguishing 2 mm edges in Figure 5 in which arrows indicate the putative afferents that are included or excluded from the population response, and suggest that subjects might be able to distinguish the two orientations by recognizing which fibers are active, and which are silent. This is grossly oversimplified, because as two of the authors (Pruszynski and Johansson) have shown, individual afferents are able to summate responses from simultaneous stimulation of sensory branches, while others might not reach the spike threshold. The model also seems to assume all-or-none output from the individual sensory terminals. The model might be particularly useful for distinguishing the edge length effects, rather than the angular orientation of the edge. The best solution would be for the authors to present a formal description of their model, including equations used to distinguish orientations, rather than the simple graphics used in Figure 5. (Also, a minor note, the descriptor "toy model" is still used in the Figure legend; please correct this.).

Second, the hyperacuity model with connected subfields is proposed to explain the behavioral data. It is an intriguing model and it does outperform the traditional model with uniform receptive fields. However, this physiology-inspired model should be applicable for the active tasks as well as the traditional passive tasks (i.e., the model works regardless of active exploration along the skin). Thus it is still not clear how the model helps to resolve the discrepancy between the current findings (superior tactile acuity) and the previous finding (poor acuity). Moreover, the model does not relate to the hypothetical perception-action account that is favored by the authors. The authors need to better situate their model into the perception-action framework they use to argue the superiority of tactile acuity.

2) Active vs. Passive: The most important result of the paper is the impressive tactile acuity in the pointer-alignment task when compared with the poor performance reported in previous perceptual tasks. As was pointed out in the first review, this improvement is possibly a result of task difference, especially because the current task is an active touch task. In comparison, those passive perceptual estimation tasks typically put an edge stimulus perpendicularly against the skin (e.g., Peters, Staibano and Goldreich, 2015, which is also quoted by the authors). The large performance improvement by active touch has been known for nearly a century; for example, as pointed out by David Katz in 1925: "The full richness of the palpable world is opened up to the touch organ only through movements." Yet the authors basically discount this possibility in the Discussion: "However, provided that the skin deformations are similar to each other under active and passive conditions, active information seeking seems not to significantly improve spatial discrimination in perceptual tactile tasks (Lamb, 1983; Lederman, 1981; Vega-Bermudez, Johnson and Hsiao, 1991)." Interesting, the studies cited do not use the same passive conditions mentioned in the sentence. These "passive" conditions move the stimulus against an immobilized finger, as opposed to simply push an edge stimulus against the skin. Thus, when the authors wanted to show the superior performance they compared their active task against one type of "passive" task; when they tried to show that active information seeking does not improve edge discrimination, they quoted another type of "passive" task.

It seems clear that active touch is the main reason behind the superior tactile acuity reported here. The authors argue that "since the edges deformed the skin essentially through perpendicular skin indentation both in our task and in the perceptual tasks, it is unclear whether active touch contributed to the higher edge orientation sensitivity in our study." However, the pointer-alignment task not only has a perpendicular skin indentation, it also creates (at least) some sort of shear force when the finger moves the edge. This is a type of important sensory feedback for tactile processing of edge orientation, which should not be neglected.

The authors favor an alternative theory that tactile information is spatially processed in different ways for active tasks than for perceptual estimation tasks: "A third factor, which in our view may be most important for superior edge orientation sensitivity in object manipulation compared to perceptual tasks, concerns differences in how tactile information is spatially processed to support the behaviour in the two situations. In object manipulation, tactile information about edge orientation is naturally mapped onto the orientation of an object in external space and hence in the same space as the task goal. Moreover, because the object is mechanically coupled to the hand, the spatial transformation required to complete the task (i.e., object rotation) can be directly mapped onto motor commands." However, it is not clear how tactile information about edge orientation is naturally mapped onto the orientation of an object in external space for object manipulation, but not for perceptual estimation nor why spatial transformation (what exactly it is?) can be directly mapped onto motor commands can support superior tactile performance. The authors imply that their account is similar to the two-visual streams theory. However, to make this link, they need to prove or assume that the bottom-up tactile feedback is the same for the active and passive tasks. Since their pointer-alignment task involves very different tactile feedback than those perceptual estimation tasks, it is too early to make this link.

Thus, as with the second concern outlined in the critique of the model, the authors need to much better lay out their case as to the plausible mechanisms by which active movements improve this fine tactile acuity.

3) Task Description: The description of the task protocols remains opaque or misleading. For example, the authors categorize their task as a two-alternative forced choice protocol (presumably meaning that the subjects can choose to rotate the edge clockwise or counterclockwise), but as noted in the expanded Materials and methods: "A raised edge, centered on the plate [rest position of the hand during the intertrial interval] and spanning its entire length, was pointing towards the target position (i.e., 0°). […] The function of this edge was to offer the participants a tactile reference for the finger's home position." This means that the protocol is really a match-to-sample task, in which the test edge on each trial is compared to a standard edge that is easily matched when testing with the "infinite" edge. This distinction is important because the authors refer to spatial "working memory" in the main text [see subsection “Action versus perception”, second paragraph where the term is raised four times], in terms of the orientation on the previous trial: ".… direct sensorimotor information (visual, proprioceptive, efference) about the target position was last available ~3 s before the touch, i.e., at the end of the previous trial." This reference to working memory of earlier trials again appears in the subsection “Tactile versus visual acuity”, and” Neural mechanisms”, first paragraph. The subjects are presumably using the information learned during the intertrial interval when their finger rests on an "infinite edge" of the appropriate orientation.

At a minimum the authors need to specify in the main description of the task, in the first paragraph of the Results, that they have used a match-to-sample task, in which the sample is presented during the home position in the intertrial interval, and the subject needs to match the orientation of the test edge to that of the sample edge. Referring to memory of test edges on previous trials is inappropriate and indeed incorrect. Linking the test edge to that of the visual pointer used for feedback distorts what actually is needed for performance, although it may aid the subject in visualizing what is meant by orientation. All readers should be aware of the fact that the tactile cues were provided during the intertrial interval in the Results and Discussion, rather than being buried in the Materials and methods, as this is a critical feature of the task design that gets overlooked when interpreting the results.

[Editors' note: further revisions were requested prior to acceptance, as described below.]

Thank you for submitting your article "Fast and accurate edge orientation processing during object manipulation" for consideration by *eLife*. Your article has been reviewed by Sabine Kastner as the Senior Editor and a Reviewing Editor.

The after reviewing the new revisions, Reviewing Editor has drafted this decision to help you prepare a revised submission.

While the second revision of this manuscript has gone a long way to addressing the concerns laid out in the review process, there are lingering concerns regarding the descriptions of the model itself. The details provided in the revised Materials and methods section do not provide enough information for someone to be able to recreate the model themselves. It is advised that the following changes be made.

1) Provide an algorithmic description of the model in the form of an algorithm table.

2) Provide clear functional forms of details like activation functions, receptive field distribution, receptive field size, etc.

If the authors require clarification, they are more than welcome to reach out to the Reviewing Editor (Timothy Verstynen) directly for details.

---

## [Author Response]

Essential revisions:1) The Model: The toy model presented at the end of the paper is somewhat fundamental for interpreting the findings in a physiological context. As reviewer #3 points out, it is a "simple" model, not a "toy" model, and has substantial experimental support from the authors' own work, and that of other investigators. It is both out of place being in the Discussion section and lacking clear methodological details. Please move the model into the core of the paper, with adequate descriptions in the Materials and methods on how it is implemented and a more complete description of the findings in the Results. The authors are encouraged to look at recent study from Ellen Lumpkin's lab on summated responses from Merkel cell afferents when stimulated together (Lesniak et al., 2014), and should also look at Suresh et al., 2016 as these findings may bolster the biological validity of the model being proposed.

As suggested by the reviewers, we have moved the model into the Results (section “A simple model of edge orientation processing”) and provided the necessary methodological details in the Materials and methods (section “Model”). The suggested papers are now cited either in the Introduction or the Discussion, or in both those sections.

2) Active vs. Passive: The main novel finding of the paper is the fast and accurate tactile capacity discovered in the new paradigm; however, this discrepancy with existing literature on tactile perception is likely due to task differences. Reviewers #1 and #2 pointed out that previous research in tactile discrimination and identification typically used passive paradigms, i.e., stimuli were applied on an immobilized finger pad. Even previous animal studies and their associated theoretical models utilized anesthetized monkeys (e.g., Friedman et al., 2002). The present study required the participant to actively touch the edge and then moved it without any constraints. The authors also noted that their task had continuous tactile inputs while previous studies briefly presented the stimuli to the fingertip before the perceptual report. Is this then somewhat an apples-and-oranges comparison? Active exploration is known to produce better perceptual performance in the other perceptual systems. Reviewer #3 pointed out that the increased sensitivity to edge orientation may reflect the pragmatic utilization of tactile signals by the motor system, rather than the cognitive utilization of matching tactile and visual edges as has been performed in previous psychophysical studies.The authors should tone down the interpretation the implications of their findings in this regard. It is not unexpected that active exploration improves sensitivity, but what is striking is the degree to which it improves tactile sensitivity. The authors should revise how they link their results to this previous passive literature to acknowledge this difference and comment on the use of sensorimotor parameters rather than verbal responses in their perceptual measurements.

As suggested by the reviewers, we have now tempered the text in the Results (section “Tactile edge orientation is extracted ad processed very quickly in manipulation”) and have added substantial text throughout the Discussion to ensure our results are properly linked to previous studies. Specifically, under the heading “Action versus perception” in the Discussion section, we have addressed various issues related to the relationship between our results and the existing literature on perception of edge orientation.

3) Measures of "Processing Speed": There was unanimous concern about the measures of processing speed reported here. If rotation onset time and time of contact force increase are measures of processing speed, wouldn't you expect them to get longer as available sensory information decreases (i.e., as edge length decreases)? Unlike perceptual reports, the time of starting to rotate the surface does not necessarily equate to the time that participants make up their mind about the edge direction. As this is an active task, participants could very likely receive more information during rotation since fast-adapting tactile endings (Meissner and Pacini endings) are better recruited during movements. This possibility is highlighted by the fact that 44% of all trials had sub-movements, which might relate to the fact that participants were adjusting their initial "judgment" based on incoming new feedback. Reviewer #2 pointed out that this possibility cannot be ruled out by ANCOVA results as reported. For this analysis, absolute alignment error was the dependent variable, rotation onset time was the covariate, and edge length and initial dial orientation were categorical variables. There was no significant correlation between absolute error and rotation onset time. Yes, onset time appears to not affect absolute error, but it is still possible that the performance was determined by perceptual inputs during the whole trial duration, no matter where the onset happened. This is perhaps one of the most critical problems in the paper. While none of the reviewers thought that this required collecting new data with a better measure of processing speed, the authors should acknowledge this limitation throughout the manuscript (not just as an addition to the Discussion) and temper their descriptions of processing speed per se.

As suggested by the reviewers, we have now modified text in the Results and added substantial text in the Discussion to more precisely delineate the inferences we can make about processing speed based on our data. This matter is addressed in the Discussion section entitled “Processing speed”. We have also added additional results (section “Tactile edge orientation is extracted and processed very quickly in manipulation”, fifth paragraph) that account for the whole trial duration.

4) Visual vs. Tactile: There were also several concerns about the clarity of the links between the tactile experiment and the visual experiment. All three reviewers felt that the motivation, description, and interpretation of this across modality comparison needed more detail. Reviewer #1 felt that the authors were downplaying an apparent consistent difference between the modalities. The authors try to say that performance was essentially identical across the tactile and visual conditions, however, Figure 4B and Figure 3—figure supplement 1B and C clearly show a consistent within-subject difference across conditions (with the mean trends masked by the between subject variance). So it appears that sensory modality does matter. Reviewer #2 pointed out that the tactile only and visual only experiments didn't just vary in their perceptual modality because participants received visual inputs until they touched the contact surface. Thus, the perceptual judgment was affected by memory decay of the visual representation of the dial. In contrast, in the main experiment, tactile feedback was continuously available for guiding the movement. Thus, while two experiments achieved similar performance, it does not necessarily mean that this non-effect is due to comparable unimodal performance as claimed by the authors. It is possibly due to the availability of sensory inputs too. The authors correctly pointed out that tactile stimuli were continuously available when they discussed the difference between their study and those early tactile perception studies. However, this discontinuous-versus-continuous distinction is also present in their comparison between the visual and tactile experiments. This is perhaps the most critical problem identified in the study. Had the authors made matching tactile and visual sensitivities be the critical focus of the study, these concerns would have been seen as a fundamental/critical flaw requiring new data. However, since the main focus is on describing the sensitivity of tactile perception during active exploration (and the comparison to visual sensitivity gives context), we suggest that the authors identify either analytical solutions to address these concerns and/or highlight these limitations and temper the interpretive links between tactile and visual modalities.

As suggested by the reviewers, we now explicitly highlight the limitations of directly comparing the visual and tactile tasks in the Discussion section entitled “Tactile versus visual acuity”.

5) Task Description: The task description itself was found to be opaque, particularly for the broad/non-specialized audience of eLife. Reviewer #3 points out that this motor task is similar to adjusting the minute hand on a watch to the 12:00 position from either the 11:55 or 12:05 position. In the current task, the subject touches the test edge and uses the tactile sense to detect its orientation, and then turn the dial in the opposite direction [clockwise or counterclockwise] to the upright position. In other words, the subject must recognize the edge orientation, and make an equal and opposite hand movement to accomplish the task goals. The task instructions do not explicitly tell the subject to detect the initial angle of the edge, only that the dial needs to be turned to the upright (i.e., vertical) target position. Edge orientations studied were +/- 10, 20, or 30°; subjects were required to move the dial to within 2° of the vertical position. While the task is clearly described in the extended Materials and methods, the presentation needs a better descriptor than in the current version. The authors are encouraged to use the clock face metaphor or that of a compass needle rotated from NE or NW to due North in order to make the details more accessible to a broader audience. The Reviewing Editor notes that the authors may choose to use another technique to increase accessibility of the task descriptions.

This is a great suggestion. We have adopted the compass analogy at the beginning of the Results to make it easier to penetrate by the general readership of *eLife*.

[Editors' note: further revisions were requested prior to acceptance, as described below.]

The open issues from items in the original summary are as follows:1) The Model: All reviewers thought that the relocation of the model into the Results and expanded description have helped better integrate this with the main psychophysical findings. However, there remain two points of concern with this model.The first concern centers on clarity of the revised description. The authors note that cutaneous mechanoreceptors in the human hand branch at their distal endings, and that SA1 and RA1 afferents are capable of spike generation at any and all of these branches. They propose that the receptive field sensitivity reflects the stimulation pattern of any and all of these branches. This is a very intriguing idea, but the model is not sufficiently described in the report. The basic idea is that subjects can distinguish whether particular afferents are included or excluded if the parent axon fires a spike. They illustrate a plausible mechanism for distinguishing 2 mm edges in Figure 5 in which arrows indicate the putative afferents that are included or excluded from the population response, and suggest that subjects might be able to distinguish the two orientations by recognizing which fibers are active, and which are silent. This is grossly oversimplified, because as two of the authors (Pruszynski and Johansson) have shown, individual afferents are able to summate responses from simultaneous stimulation of sensory branches, while others might not reach the spike threshold. The model also seems to assume all-or-none output from the individual sensory terminals. The model might be particularly useful for distinguishing the edge length effects, rather than the angular orientation of the edge. The best solution would be for the authors to present a formal description of their model, including equations used to distinguish orientations, rather than the simple graphics used in Figure 5. (Also, a minor note, the descriptor "toy model" is still used in the Figure legend; please correct this.).

As requested, we have expanded the description of the model in the Materials and methods and now provide the few key equations defining activation of each synthetic unit (Equation 1), the activation of each receptor element (Equations 2A and 2B) and the normalization procedure which factors into discrimination threshold (Equation 3). As requested, we have also removed the word “toy” from the legend in Figure 5 and searched the document to ensure that this word does not appear elsewhere.

With respect to our choices in terms of simplifying the neurobiology, these are now detailed more clearly in the Materials and methods, Results and Discussion. First, as was previously described in the Discussion, it is important to emphasize that other papers (e.g. Wheat et al., 1995) have looked at intensity-based codes and have shown that, under some assumptions, that these can provide an explanation for the kind of acuity we report. This is now mentioned immediately in the Results (section “A simple model of edge orientation processing”, first paragraph) and is re-emphasized in the Discussion (section “Neural mechanisms”, first paragraph).

Briefly, previously proposed models indicate that spatial details are resolved based on the relative discharge rates of first-order tactile neurons having partly overlapping receptive fields with smooth Gaussian-like sensitivity profiles (see Friedman et al., 2002; Dodson et al., 1998; Wheat, Goodwin and Browning 1995; Saal et al., 2017; Loomis and Collins 1978; Khalsa et al., 1998). In such a scenario, the relative firing rate of some population of stimulated neurons (estimated over some epoch of time) could theoretically provide infinite resolution about the spatial features of a stimulus. However, the devil is in the details of how much noise is present, the precise geometry of the receptive fields, the temporal epoch available for estimating firing rates and the amount of overlap.

Our focus is on another edge case, the ‘pure’ coincidence of spiking activity, as another possible alternative code for such functions. We motivate our choice, now explained in the Results (subsection “A simple model of edge orientation processing”) by (a) the observation in Pruszynski and Johansson (2014) that spike timing information generated by single neurons (the basis of a coincidence code) is a better source of edge orientation information than changes in intensity, (b) that codes relying on spike timing may be better suited for quickly signalling information for the purposes of control (e.g. Johansson and Birzeniks, 2004), and (c) because there exist reasonable decoding mechanisms that could capture such information upstream (as cited in the paper).

Of course, neither our present work nor any other previous study can specifically identify which code is actually being used, a point we now make in the Discussion (subsection “Neural mechanisms”, second paragraph). Moreover, we point to recent methodological developments enabling large-scale recordings from second-order tactile neurons in the brainstem of awake monkeys (Suresh et al., 2017), an exciting development that provides a means to directly investigate such questions in future.

Second, the hyperacuity model with connected subfields is proposed to explain the behavioral data. It is an intriguing model and it does outperform the traditional model with uniform receptive fields. However, this physiology-inspired model should be applicable for the active tasks as well as the traditional passive tasks (i.e., the model works regardless of active exploration along the skin). Thus it is still not clear how the model helps to resolve the discrepancy between the current findings (superior tactile acuity) and the previous finding (poor acuity). Moreover, the model does not relate to the hypothetical perception-action account that is favored by the authors. The authors need to better situate their model into the perception-action framework they use to argue the superiority of tactile acuity.

As described above, the model is not intended to (nor can it or any other model that we are aware of) provide a specific reason for why tactile acuity during hand control is better than during passive perceptual studies. The reasons for such differences are complex and multi-factorial and far beyond anyone’s present understanding of the tactile processing pathway (see reviewer comments our response below, and our extended Discussion section, entitled “Action versus Perception”). The goal of our modeling effort, as we now make clear in the first paragraph of the Results (see above) and Discussion (see above), is to show that there exists an alternative peripheral coding scheme to the firing rate interpolation scheme previously proposed, whereby coincidence of activation across neurons could provide (i) the requisite spatial resolution observed in our experimental and, equally important, (ii) the short processing time present in object manipulation. And, in the context of this model, we show that for such a coincidence code to work well, the system needs to leverage spatially complex receptive fields of the type we have identified in the glabrous skin of the hand (Figure 5c and subsection “A simple model of edge orientation processing”, last paragraph).

2) Active vs. Passive: The most important result of the paper is the impressive tactile acuity in the pointer-alignment task when compared with the poor performance reported in previous perceptual tasks. As was pointed out in the first review, this improvement is possibly a result of task difference, especially because the current task is an active touch task. In comparison, those passive perceptual estimation tasks typically put an edge stimulus perpendicularly against the skin (e.g., Peters, Staibano and Goldreich, 2015, which is also quoted by the authors). The large performance improvement by active touch has been known for nearly a century; for example, as pointed out by David Katz in 1925: "The full richness of the palpable world is opened up to the touch organ only through movements." Yet the authors basically discount this possibility in the Discussion: "However, provided that the skin deformations are similar to each other under active and passive conditions, active information seeking seems not to significantly improve spatial discrimination in perceptual tactile tasks (Lamb, 1983; Lederman, 1981; Vega-Bermudez, Johnson and Hsiao, 1991)." Interesting, the studies cited do not use the same passive conditions mentioned in the sentence. These "passive" conditions move the stimulus against an immobilized finger, as opposed to simply push an edge stimulus against the skin. Thus, when the authors wanted to show the superior performance they compared their active task against one type of "passive" task; when they tried to show that active information seeking does not improve edge discrimination, they quoted another type of "passive" task.It seems clear that active touch is the main reason behind the superior tactile acuity reported here. The authors argue that "since the edges deformed the skin essentially through perpendicular skin indentation both in our task and in the perceptual tasks, it is unclear whether active touch contributed to the higher edge orientation sensitivity in our study." However, the pointer-alignment task not only has a perpendicular skin indentation, it also creates (at least) some sort of shear force when the finger moves the edge. This is a type of important sensory feedback for tactile processing of edge orientation, which should not be neglected.

We did not intend to discount the possibility that the high acuity we observe arises because our task involves actively contacting, and then rotating, the contact surface. We have rewritten this section of the Discussion (entitled “Action versus perception”) to provide a more balanced perspective. Specifically, we now recognize that the contact dynamics that take place when the digit contacts with the surface may be somewhat different than the dynamics that would take place if this contact occurred passively. Second, although we have demonstrated that participants rapidly, and with high acuity, extract information about orientation prior to object rotation, we also recognize that some additional information about orientation could be extracted early during the rotation and that this information could be used to adjust the ongoing motor commands generating the rotation.

However, we disagree that it is “clear that active touch is the main reason behind the superior tactile acuity”. In our view, there are a number of possible explanations for the higher acuity observed in our task, ranging from low-level factors (e.g., contact mechanics, see above) to high-level considerations (e.g., memory interference; spatial transformations; task and context-dependent sensory processing). As noted above, we have tried to be more balanced when considering possible factors that may contribute to the differences in reported acuity across tasks while acknowledging that our present results cannot reveal a precise reason. We also emphasize that our present work cannot explain the range of acuity findings. Importantly, our study was not primarily aimed at explaining reasons for differences in how the nervous system expresses edge orientation sensitivity during object manipulation compared with perceptual tasks. Its purpose was to examine, for the first time, the accuracy and speed with which the neural system extracts and expresses tactile edge orientation information during object manipulation tasks that require fine manual dexterity. In this context, we consider that our paper makes an important contribution by demonstrating that tactile processing of edge orientation can be extremely good (i.e., fast and accurate), thus setting a higher bar for models of tactile processing and motivating new lines of empirical work to revolve the behavioural and neural coding differences.

The authors favor an alternative theory that tactile information is spatially processed in different ways for active tasks than for perceptual estimation tasks: "A third factor, which in our view may be most important for superior edge orientation sensitivity in object manipulation compared to perceptual tasks, concerns differences in how tactile information is spatially processed to support the behaviour in the two situations. In object manipulation, tactile information about edge orientation is naturally mapped onto the orientation of an object in external space and hence in the same space as the task goal. Moreover, because the object is mechanically coupled to the hand, the spatial transformation required to complete the task (i.e., object rotation) can be directly mapped onto motor commands." However, it is not clear how tactile information about edge orientation is naturally mapped onto the orientation of an object in external space for object manipulation, but not for perceptual estimation nor why spatial transformation (what exactly it is?) can be directly mapped onto motor commands can support superior tactile performance. The authors imply that their account is similar to the two-visual streams theory. However, to make this link, they need to prove or assume that the bottom-up tactile feedback is the same for the active and passive tasks. Since their pointer-alignment task involves very different tactile feedback than those perceptual estimation tasks, it is too early to make this link.Thus, as with the second concern outlined in the critique of the model, the authors need to much better lay out their case as to the plausible mechanisms by which active movements improve this fine tactile acuity.

We appreciate this comment and agree that we need to more clearly articulate our thinking on this third factor. As described in detail above, we have rewritten this entire section (“Action versus perception”) and tried to more clearly break down the differences between our task and various previous tasks and indicate how these differences may influence acuity. Based on the reviewer comments, we no longer favour a particular view but rather simply consider the possible relevant factors and ensure that the reader understands that our present work cannot disambiguate between them.

3) Task Description: The description of the task protocols remains opaque or misleading. For example, the authors categorize their task as a two-alternative forced choice protocol (presumably meaning that the subjects can choose to rotate the edge clockwise or counterclockwise), but as noted in the expanded Materials and methods: "A raised edge, centered on the plate [rest position of the hand during the intertrial interval] and spanning its entire length, was pointing towards the target position (i.e., 0°). […] The function of this edge was to offer the participants a tactile reference for the finger's home position." This means that the protocol is really a match-to-sample task, in which the test edge on each trial is compared to a standard edge that is easily matched when testing with the "infinite" edge. This distinction is important because the authors refer to spatial "working memory" in the main text [see subsection “Action versus perception”, second paragraph where the term is raised four times], in terms of the orientation on the previous trial: ".… direct sensorimotor information (visual, proprioceptive, efference) about the target position was last available ~3 s before the touch, i.e., at the end of the previous trial." This reference to working memory of earlier trials again appears in the subsection “Tactile versus visual acuity”, and “Neural mechanisms”, first paragraph. The subjects are presumably using the information learned during the intertrial interval when their finger rests on an "infinite edge" of the appropriate orientation.At a minimum the authors need to specify in the main description of the task, in the first paragraph of the Results, that they have used a match-to-sample task, in which the sample is presented during the home position in the intertrial interval, and the subject needs to match the orientation of the test edge to that of the sample edge. Referring to memory of test edges on previous trials is inappropriate and indeed incorrect. Linking the test edge to that of the visual pointer used for feedback distorts what actually is needed for performance, although it may aid the subject in visualizing what is meant by orientation. All readers should be aware of the fact that the tactile cues were provided during the intertrial interval in the Results and Discussion, rather than being buried in the Materials and methods, as this is a critical feature of the task design that gets overlooked when interpreting the results.

We really appreciate this feedback; it was an oversight on our part. We agree with the reviewer and have now added a more detailed description in the Introduction to ensure that the reader is aware of this aspect of the experimental design. Moreover, we have revised the associated Discussion section (“Action versus perception”, see above comments for details; and the aforementioned section specifically to provide a more balanced description of the possible factors that differentiate previous perceptual studies and our present motor control task. Lastly, we now provide a video that explicitly shows the task being performed from a few vantage points (Results, first paragraph, subsection “General Procedure” Video 1).